# FedLAP-DP: Federated Learning by Sharing Differentially Private Loss Approximations

## Abstract

This work proposes FedLAP-DP, a novel privacy-preserving approach for federated learning. Unlike previous linear point-wise gradient-sharing schemes, such as FedAvg, our formulation enables a type of global optimization by leveraging synthetic samples received from clients. These synthetic samples, serving as loss surrogates, approximate local loss landscapes by simulating the utility of real images within a local region. We additionally introduce an approach to measure effective approximation regions reflecting the quality of the approximation. Therefore, the server can recover an approximation of the global loss landscape and optimize the model globally. Moreover, motivated by the emerging privacy concerns, we demonstrate that our approach seamlessly works with record-level differential privacy (DP), granting theoretical privacy guarantees for every data record on the clients. Extensive results validate the efficacy of our formulation on various datasets with highly skewed distributions. Our method consistently improves over the baselines, especially considering highly skewed distributions and noisy gradients due to DP. The source code and setup will be released upon publication.

## 1 Introduction

Federated Learning (FL) (McMahan et al., 2017) is a distributed learning framework that allows participants to train a model collaboratively without sharing their data. Predominantly, existing works (McMahan et al., 2017; Karimireddy et al., 2020; Li et al., 2020) achieve this by training local models on clients' private datasets and sharing only the gradients with the central server. Despite extensive research over the past few years, these prevalent gradient-based methods still suffer from several challenges (Kairouz et al., 2021), such as data heterogeneity, potential risks of privacy breaches, and high communication costs.

Data heterogeneity (Hsu et al., 2019; Li et al., 2019; Karimireddy et al., 2020) often hurts performance and convergence speed. The fundamental reason is that *the local updates based on the private datasets optimize the models for their local minima but tend to be sub-optimal to the global objective*. In this work, we propose FedLAP-DP, a novel differentially private framework designed to approximate local loss landscapes and counteract biased federated optimization through the utilization of synthetic samples. As illustrated in Fig. 1, unlike the traditional gradient-sharing scheme (McMahan et al., 2017) which is prone to inherently biased global update directions, our framework transmits synthetic samples encoding the local optimization landscapes. This enables the server to faithfully reconstruct the global loss landscape, overcoming the biases incurred by conventional gradient-sharing schemes and resulting in substantial improvements in convergence speed (refer to Sec. 5). Additionally, we introduce the usage of a trusted region to faithfully reflect the approximation quality, further mitigating bias stemming from potential imperfections in the local approximation within our scheme.

Privacy protection is another crucial aspect of FL. Various studies have uncovered vulnerabilities in existing federated systems, including the risk of data leakage (Hitaj et al., 2017; Bhowmick et al., 2018; Geiping et al., 2020) and membership inference (Nasr et al., 2019; Melis et al., 2019). In response to these concerns, our framework strategically incorporates record-level differential privacy, thereby ensuring rigorous privacy guarantees for each individual data record within the system. Built upon differentially private loss approximation, our method provides reliable utility under privacy-preserving settings, especially when considering low privacy budgets.

Lastly, our framework is more communication efficient as it enables the execution of multiple optimization rounds on the server side, and – particularly for large models – transferring synthetic data is less costly than gradients in each round. We summarize our contributions as follows.

- We propose FedLAP-DP that uses synthetic images to comprehensively approach federated optimization, which has yet to be thoroughly investigated before.
- We demonstrate how to accurately identify local approximation and effective regions on clients, enabling efficient federated optimization on the server.
- FedLAP-DP delivers stringent record-level DP assurances, maintaining utility and outperforming gradient-sharing counterparts in privacy-preserving settings.
- Extensive experiments confirm that our formulation presents superior performance and convergence speed over existing gradient-sharing baselines, even with highly skewed data distributions.

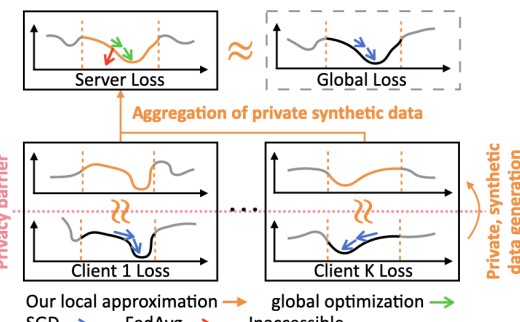

Figure 1: An overview of FedLAP-DP. It approximates local neighborhoods with synthetic images on the clients (local approximation, Sec. 4.2) and optimizes the model according to the reconstructed loss landscape on the server (global optimization, Sec. 4.3). Differential privacy is integrated to introduce privacy barriers (Sec. 4.4).

## 2  RELATED WORK

**Non-IID Data in Federated Learning** causes major challenges in FL (Kairouz et al., 2021). Existing efforts mainly fall into the following categories: variance-reduction techniques (Karimireddy et al., 2020; Yu et al., 2019), constraining the dissimilarity between clients' updates (Li et al., 2020; 2021a), and adjusting the global model to a personalized version at the inference stage (Luo et al., 2021; Li et al., 2021b; Fallah et al., 2020). In contrast, we present a novel scheme that considers valid approximation regions and effectively resolves non-IID federated optimization. Notably, a concurrent work FedDM (Xiong et al., 2023) shares a similar idea in approximating loss landscapes but employs class-wise feature matching. Despite easier synthesis, it neglects the importance of approximation quality and might introduce additional privacy risks due to class-wise optimization.

**Dataset Distillation** Our work is largely motivated by recent progress in distilling the necessary knowledge of model training into a small set of synthetic samples (Wang et al., 2018; Zhao et al., 2021; Zhao & Bilen, 2023; Cazenavette et al., 2022). Our approach is built on top of DSC (Zhao et al., 2021) with several key differences. The proposed FedLAP-DP (i) focuses on finding local approximation and assembling the global loss landscape to facilitate federated optimization, (ii) is class-agnostic and complements record-level differential privacy while prior works often consider class-wise alignment and could cause privacy risks, and (iii) is designed for multi-round training with several critical design choices. The most relevant work is Chen et al. (2022). They also consider class-agnostic distillation and differential privacy while focusing on one-round distillation rather than multiple-round federated learning.

## 3  BACKGROUND

### 3.1  FEDERATED LEARNING

In federated learning, we consider training a model $\mathbf{w}$ that maps the input $\boldsymbol{x}$ to the output prediction $y$. We assume $K$ clients participate in the training, and each owns a private dataset $\mathcal{D}_k$ with distribution $p_k$. We use the subscript $k$ to represent the indices of clients, and the superscript $m$ and $t$ to denote the $m$-th communication round and $t$-th local step, respectively, unless stated otherwise.

Overall, the learning objective is to find the optimal model $\mathbf{w}$ that minimizes the empirical global loss over the population distribution:

$$\mathcal{L}(\mathbf{w}) = \mathbb{E}_{(\boldsymbol{x},y)\sim p}[\ell(\mathbf{w}, \boldsymbol{x}, y)] = \frac{1}{N}\sum_{j=1}^{N}\ell(\mathbf{w}, \boldsymbol{x}_j, y_j) \tag{1}$$

where $\ell$ could be arbitrary loss criteria such as cross-entropy and $N$ is the total dataset size. However, in federated settings, direct access to the global objective is prohibited as all client data is stored locally. Instead, the optimization is conducted on local surrogate objectives $\mathcal{L}_k(\mathbf{w})$:

$$\mathcal{L}(\mathbf{w}) = \sum_{k=1}^{K}\frac{N_k}{N}\mathcal{L}_k(\mathbf{w})\,,\; \mathcal{L}_k(\mathbf{w}) = \sum_{j=1}^{N_k}\frac{1}{N_k}\ell(\mathbf{w}, \boldsymbol{x}_j, y_j),$$

where $(\boldsymbol{x}_j, y_j)$ are data samples from the client dataset $\mathcal{D}_k$.

Existing methods, such as FedAvg (McMahan et al., 2017), simulate stochastic gradient descent on the global objective by performing local gradient updates and periodically aggregating and synchronizing them on the server side. Specifically, at the $m$-th communication round, the server broadcasts the current global model weights $\mathbf{w}_g^{m,1}$ to each client, who then performs $T$ local iterations with learning rate $\eta$.

$$\mathbf{w}_k^{m,1} \leftarrow \mathbf{w}_g^{m,1}, \forall k \in [K] \quad \text{and} \quad \mathbf{w}_k^{m,t+1} = \mathbf{w}_k^{m,t} - \eta\nabla\mathcal{L}_k(\mathbf{w}_k^{m,t}), \forall t \in [T] \tag{2}$$

The local updates $\Delta\mathbf{w}_k^m$ are then sent back to the server and combined to construct $\widehat{\boldsymbol{g}^m}$, a linear approximation of the true global update $\boldsymbol{g}^m$:

$$\widehat{\boldsymbol{g}^m} = \sum_{k=1}^{K}\frac{N_k}{N}\Delta\mathbf{w}_k^m = \sum_{k=1}^{K}\frac{N_k}{N}(\mathbf{w}_k^{m,T} - \mathbf{w}_k^{m,1}) \quad \text{and} \quad \mathbf{w}_g^{m+1,1} = \mathbf{w}_g^{m,1} - \eta\widehat{\boldsymbol{g}^m} \tag{3}$$

In this work, we focus on the conventional FL setting in which each client retains their private data locally, and the local data cannot be directly accessed by the server. Every data sample is deemed private, with neither the server nor the clients using any additional (public) data, which stands in contrast to previous works that require extra data on the server side (Zhao et al., 2018; Li & Wang, 2019). Furthermore, all clients aim towards a singular global objective (Eq. 1), which is distinct from personalized approaches wherein evaluations are conducted based on each client's unique objective and their own data distribution (Li et al., 2021b; Fallah et al., 2020).

## 3.2 Non-IID Challenges

The heterogeneity of client data distributions presents several major challenges to FL, such as a significant decrease in the convergence speed (and even divergence) and the final performance when compared to the standard IID training setting (Khaled et al., 2019; Li et al., 2020; Karimireddy et al., 2020; Li et al., 2019). This can be easily seen from the mismatch between the local objectives that are being solved and the global objective that we are indeed aiming for, i.e., $\mathcal{L}_k(\mathbf{w}) \neq \mathbb{E}_{(\boldsymbol{x},y)\sim p}[\ell(\mathbf{w}, \boldsymbol{x}, y)]$ if $p_k \neq p$ for some k. Executing multiple local steps on the local objective (Eq. 2) makes the local update $\Delta\mathbf{w}_k^m$ deviate heavily from the true global gradient $\nabla\mathcal{L}(\mathbf{w})$, inevitably resulting in a biased approximation of the global gradient via Eq. 3, i.e., $\widehat{\boldsymbol{g}^m} \neq \boldsymbol{g}^m$, where $\boldsymbol{g}^m$ is derived from the true loss $\mathcal{L}(\boldsymbol{w})$ (See Fig. 1 for a demonstration.).

Despite significant advances achieved by existing works in alleviating divergence issues, these methods still exhibit bias towards optimizing the global objective as they rely on the submitted client updates $\Delta\mathbf{w}_k^m$, which only indicate the direction towards the client's local optimum.

In contrast, our method communicates the synthetic samples $\mathcal{S}_k$ that encode the local optimization landscapes, i.e., gradient directions within a trust region around the starting point and summarize on possible trajectories $(\mathbf{w}_k^{m,1}, \mathbf{w}_k^{m,2}, ..., \mathbf{w}_k^{m,T+1})$, as opposed to existing methods that communicate a single direction $\Delta\mathbf{w}_k^m = \mathbf{w}_k^{m,T+1} - \mathbf{w}_k^{m,1}$. This fundamental change provides the central server with a global perspective that faithfully approximates the ground-truth global optimization (See Fig. 1 top row) than existing approaches.

### 3.3 DIFFERENTIAL PRIVACY

Differential Privacy (DP) provides theoretical guarantees of privacy protection while allowing for quantitative measurement of utility. We review several definitions used in this work in this section.

**Definition 3.1** (Differential Privacy (Dwork et al., 2014))**.** A randomized mechanism $\mathcal{M}$ with range $\mathcal{R}$ satisfies $(\varepsilon, \delta)$-DP, if for any two adjacent datasets $E$ and $E'$, i.e., $E' = E \cup \{x\}$ for some $x$ in the data domain (or vice versa), and for any subset of outputs $O \subseteq \mathcal{R}$, it holds that

$$\Pr[\mathcal{M}(E) \in O] \le e^{\varepsilon} \Pr[\mathcal{M}(E') \in O] + \delta \qquad (4)$$

Intuitively, DP guarantees that an adversary, provided with the output of $\mathcal{M}$, can only make nearly identical conclusions (within an $\varepsilon$ margin with probability greater than $1 - \delta$) about any specific record, regardless of whether it was included in the input of $\mathcal{M}$ or not (Dwork et al., 2014). This suggests that, for any record owner, a privacy breach due to its participation in the dataset is unlikely.

In FL, the notion of *adjacent (neighboring) datasets* used in DP generally refers to pairs of datasets differing by either one user (*user-level* DP) or a single data point of one user (*record-level* DP). Our work focuses on the latter. While there are established methods providing record-level DP for training federated models (Truex et al., 2019; Peterson et al., 2019; Kerkouche et al., 2021), these primarily operate on the transmitted single client gradients. In contrast, our novel formulation allows efficient communication of comprehensive information, thereby circumventing biased optimization and displaying improved training stability and utility.

We use the Gaussian mechanism to upper bound privacy leakage when transmitting information from clients to the server.

**Definition 3.2.** (Gaussian Mechanism (Dwork et al., 2014)) Let $f : \mathbb{R}^n \to \mathbb{R}^d$ be an arbitrary function with sensitivity being the maximum Euclidean distance between the outputs over all adjacent datasets $E$ and $E' \in \mathcal{E}$:

$$\Delta_2 f = \max_{E, E'} \|f(E) - f(E')\|_2 \qquad (5)$$

The Gaussian Mechanism $\mathcal{M}_\sigma$, parameterized by $\sigma$, adds noise into the output, i.e.,

$$\mathcal{M}_\sigma(x) = f(x) + \mathcal{N}(0, \sigma^2 \mathbb{I}). \qquad (6)$$

$\mathcal{M}_\sigma$ is $(\varepsilon, \delta)$-DP for $\sigma \ge \sqrt{2 \ln(1.25/\delta)} \Delta_2 f / \varepsilon$.

Moreover, we use the following theorem to guarantee that the privacy leakage is bounded upon obtaining gradients from real private data in our framework. This forms the basis for the overall privacy guarantee of our framework and enables us to enhance the approximation quality without introducing additional privacy costs.

**Theorem 3.3.** *(Post-processing (Dwork et al., 2014)) If $\mathcal{M}$ satisfies $(\varepsilon, \delta)$-DP, $G \circ \mathcal{M}$ will satisfy $(\varepsilon, \delta)$-DP for any data-independent function $G$.*

## 4 FEDLAP-DP

### 4.1 OVERVIEW

Unlike existing approaches that typically communicate the local update directions to approximate the global objective (Eq. 3), we propose FedLAP to directly simulate the global optimization by transmitting a small set of synthetic samples that reflect the local loss landscapes (Fig. 1).

Let $p_k$ and the $p_{\mathcal{S}_k}$ be the distribution of the real client dataset $\mathcal{D}_k$ and the corresponding synthetic dataset $\mathcal{S}_k$, respectively. We formalize our objective and recover the global objective as follows:

$$\mathbb{E}_{(x,y) \sim p_k}[\ell(\mathbf{w}, x, y)] \simeq \mathbb{E}_{(\hat{x}, \hat{y}) \sim p_{\mathcal{S}_k}}[\ell(\mathbf{w}, \hat{x}, \hat{y})] \quad \text{and} \quad \mathcal{L}(\mathbf{w}) = \sum_{k=1}^{K} \frac{N_k}{N} \mathcal{L}_k(\mathbf{w}) \simeq \sum_{k=1}^{K} \frac{N_k}{N} \widehat{\mathcal{L}}_k(\mathbf{w}) \quad (7)$$

Thus, performing global updates is then equivalent to conducting vanilla gradient descent on the recovered global objective, i.e., by training on the synthetic set of samples.

We demonstrate our framework in Fig. 1. In every communication round, synthetic samples are optimized to approximate the client's local loss landscapes (Sec 4.2) and then transmitted to the server.

The server then performs global updates on the synthetic samples to simulate global optimization (Sec 4.3). Lastly, we show in Sec 4.4 that our method is seamlessly compatible with *record-level* differential privacy, resulting in FedLAP-DP. The overall algorithm is depicted in Algorithm 1 with the indices $i$ being the number of training trajectories observed by the synthetic images and $j$ being the number of updates on a sampled real batch. We omit the indices in the following for conciseness.

---

**Algorithm 1** FedLAP

---

**function** ServerExecute:
  **Initialize** global weight $\mathbf{w}_g^{1,1}$, radius $r$
  /* Local approximation */
  **for** $m = 1, \ldots, M$ **do**
    **for** $k = 1, \ldots, K$ **do**
      $\mathcal{S}_k, r_k \leftarrow \text{ClientsExecute}(k, r, \mathbf{w}_g^{m,1})$
    **end for**
    /* Global optimization */
    $r_g \leftarrow \min\{r_k\}_{k=1}^K$
    $t \leftarrow 1$
    **while** $\|\mathbf{w}_g^{m,1} - \mathbf{w}_g^{m,t}\| < r_g$ **do**
      $\mathbf{w}_g^{m,t+1} = \mathbf{w}_g^{m,t} - \sum_{k=1}^K \eta \frac{N_k}{N} \nabla\mathcal{L}(\mathbf{w}_g^{m,t}, \mathcal{S}_k)$
      $t \leftarrow t + 1$
    **end while**
    $\mathbf{w}_g^{m+1,1} \leftarrow \mathbf{w}_g^{m,t}$
  **end for**
  **Return:** global model weight $\mathbf{w}_g^{M+1,1}$

**function** ClientExecute($k, r, \mathbf{w}_g^{m,1}$) :
  **Initialize** $\mathcal{S}_k$: $\{\hat{\boldsymbol{x}}_k^m\}$ from Gaussian noise or $\{\hat{\boldsymbol{x}}_k^{m-1}\}$, $\{\hat{y}_k\}$ to be a balanced set
  **for** $i = 1, \ldots, R_i$ **do**
    /* Resample training trajectories */
    Reset $t \leftarrow 1$, model $\mathbf{w}_k^{m,1} \leftarrow \mathbf{w}_g^{m,1}$, and $\mathcal{S}_k^{i,0} \leftarrow \mathcal{S}_k^{i-1}$
    **while** $\|\mathbf{w}_k^{m,t} - \mathbf{w}_k^{m,1}\| < r$ **do**
      Sample real data batches $\{(\mathbf{x}_k, y_k)\}$ from $\mathcal{D}_k$
      Compute $g^{\mathcal{D}} = \nabla\mathcal{L}(\mathbf{w}_k^{m,t}, \{(\mathbf{x}_k, y_k)\})$
      **for** $j = 1, \ldots, R_b$ **do**
        /* Update synthetic set $\mathcal{S}_k$ given the real gradient */
        $\mathcal{S}_k^{i,j+1} = \mathcal{S}_k^{i,j} - \tau\nabla_{\mathcal{S}_k}\mathcal{L}_{dis}\left(g^{\mathcal{D}}, \nabla\mathcal{L}(\mathbf{w}_k^{m,t}, \mathcal{S}_k^{i,j})\right)$
      **end for**
      **for** $l = 1, \ldots, R_l$ **do**
        /* Update local models from $\mathbf{w}_k^{m,t}$ to $\mathbf{w}_k^{m,t+l}$ */
        $\mathbf{w}_k^{m,t+1} = \mathbf{w}_k^{m,t} - \eta\nabla\mathcal{L}(\mathbf{w}_k^{m,t}, \mathcal{S}_k)$
        $t \leftarrow t + 1$
      **end for**
    **end while**
  **end for**
  Measure $r_k$ on $\mathcal{D}_k$ (See Fig. 2)
  **Return:** Synthetic set $\mathcal{S}_k^{R_i}$, calibrated radius $r_k$

---

## 4.2 LOCAL APPROXIMATION

The goal of this step is to construct a set of synthetic samples $\mathcal{S}_k$ that accurately captures necessary local information for subsequent global updates. A natural approach would be to enforce similarity between the gradients obtained from the real client data and those obtained from the synthetic set:

$$\nabla_{\mathbf{w}}\mathbb{E}_{(\boldsymbol{x},y)\sim p_k}[\ell(\mathbf{w}, \boldsymbol{x}, y)] \simeq \nabla_{\mathbf{w}}\mathbb{E}_{(\hat{\boldsymbol{x}},\hat{y})\sim p_{\mathcal{S}_k}}[\ell(\mathbf{w}, \hat{\boldsymbol{x}}, \hat{y})]$$

We achieve this by minimizing the distance between the gradients:

$$\underset{\mathcal{S}_k}{\arg\min} \quad \mathcal{L}_{\text{dis}}\left(\nabla\mathcal{L}(\mathbf{w}, \mathcal{D}_k), \nabla\mathcal{L}(\mathbf{w}, \mathcal{S}_k)\right) \quad (8)$$

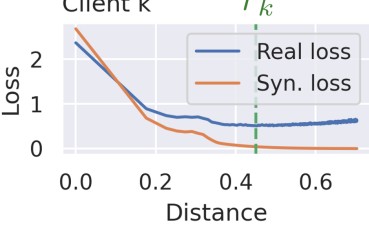

Figure 2: $r_k$ selection. The loss on private real and synthetic data decreases initially but deviates later. $r_k$ is defined as the turning points with the smallest real loss.

where $\nabla\mathcal{L}(\mathbf{w}, \mathcal{D}_k))$ denotes the stochastic gradient of network parameters on the client dataset $\mathcal{D}_k$, and $\nabla\mathcal{L}(\mathbf{w}, \mathcal{S}_k)$ the gradient on the synthetic set for brevity. $\mathcal{L}_{\text{dis}}$ can be arbitrary metric that measures the similarity. We follow Zhao et al. (2021) and adopt a layer-wised cosine distance coupled with a mean square error term to encode the directional information of the gradients and regulate the discrepancy in their magnitudes (see Sec. B and D.1 for more details and analysis).

While solving Eq. 8 for every possible $\mathbf{w}$ would lead to perfect recovery of the ground-truth global optimization in principle. However, it is practically infeasible due to the large space of (infinitely many) possible values of $\mathbf{w}$. Additionally, as $|\mathcal{S}_K|$ is set to be much smaller than $N_k$ (for the sake of communication efficiency), an exact solution may not exist, resulting in approximation error for some $\mathbf{w}$. To address this, we explicitly constrain the problem space to be the most achievable region for further global updates. Specifically, we consider $\mathbf{w}_k$ that is sufficiently close to the initial point of the local update and is located on the update trajectories (Eq. 10). Formally,

$$\underset{\mathcal{S}_k}{\arg\min} \sum_{t=1}^{T} \mathcal{L}_{\text{dis}}\left(\nabla\mathcal{L}(\mathbf{w}_k^{m,t}, \mathcal{D}_k), \nabla\mathcal{L}(\mathbf{w}_k^{m,t}, \mathcal{S}_k)\right) \quad (9)$$

$$\text{s.t.} \quad \|\mathbf{w}_k^{m,t} - \mathbf{w}_k^{m,1}\| < r \quad \text{and} \quad \mathbf{w}_k^{m,t+1} = \mathbf{w}_k^{m,t} - \eta\nabla\mathcal{L}(\mathbf{w}_k^{m,t}, \mathcal{S}_k), \quad (10)$$

where $r$ represents a radius suggested by the server, defining the coverage of update trajectories, and $\eta$ denotes the model update learning rate shared among the server and clients.

In the $m$-th communication round, the clients first synchronize the local model $\mathbf{w}_k^{m,1}$ with the global model $\mathbf{w}_g^{m,1}$ and initialize the synthetic features $\{\hat{\mathbf{x}}_k^m\}$ either from Gaussian noise or to be the ones obtained from the previous round $\{\hat{\mathbf{x}}_k^{m-1}\}$. Synthetic labels $\{\hat{y}_k\}$ are initialized to be a fixed, balanced set and are not optimized during the training process. The number of synthetic samples $|\mathcal{S}_k|$ is kept equal for all clients in our experiments, though it can be adjusted for each client depending on factors such as local dataset size and bandwidth in practice.

To simulate the local training trajectories, the clients alternate between updating synthetic features using Eq. 9 and updating the local model using Eq. 10. This process continues until the current local model weight $\mathbf{w}_k^{m,t}$ exceeds a predefined region $r$ determined by the Euclidean distance on the flattened weight vectors, meaning it is no longer close to the initial point. On the other hand, the server optimization should take into consideration the approximation quality of $\mathcal{S}_k$. Thus, as illustrated in Fig. 2, each client will suggest a radius $r_k$ indicating the distance that $\mathcal{S}_k$ can approximate best within the radius $r$. For the DP training setting, we make the choice of $r_k$ data-independent by setting it to be a constant (the same as $r$ in our experiments).

### 4.3 GLOBAL OPTIMIZATION

Once the server received the synthetic set $\mathcal{S}_k$ and the calibrated radius $r_k$, global updates can be performed by conducting gradient descent directly on the synthetic set of samples. The global objective can be recovered by $\widehat{\mathcal{L}}_k(\mathbf{w})$ according to Eq. 7 (i.e., training on the synthetic samples), while the scaling factor $\frac{N_k}{N}$ can be treated as the scaling factor of the learning rate when computing the gradients on samples from each synthetic set $\mathcal{S}_k$, namely:

$$\mathbf{w}_g^{m,t+1} = \mathbf{w}_g^{m,t} - \sum_{k=1}^{K} \eta \cdot \frac{N_k}{N} \nabla_{\mathbf{w}} \mathcal{L}(\mathbf{w}_g^{m,t}, \mathcal{S}_k) \quad \text{s.t.} \quad \|\mathbf{w}_g^{m,t} - \mathbf{w}_g^{m,1}\| \leq \min\{r_k\}_{k=1}^{K} \quad (11)$$

The constraint in Eq. 11 enforces that the global update respects the vicinity suggested by the clients, meaning updates are only made within regions where the approximation is sufficiently accurate.

### 4.4 RECORD-LEVEL DP

While federated systems offer a basic level of privacy protection, recent works identify various vulnerabilities under the existing framework, such as membership inference (Nasr et al., 2019; Melis et al., 2019). Though Dong et al. (2022) uncovers that distilled datasets may naturally introduce privacy protection, we further address possible privacy concerns that might arise during the transfer of synthetic data in our proposed method. Specifically, we rigorously limit privacy leakage by integrating record-level DP, a privacy notion widely used in FL applications. This is especially important in cross-silo scenarios, such as collaborations between hospitals, where each institution acts as a client, aiming to train a predictive model and leveraging patient data with varying distributions across different hospitals while ensuring strict privacy protection for patients.

**Threat model.** In a federated system, there can be one or multiple colluding adversaries who have access to update vectors from any party during each communication round. These adversaries may have unlimited computation power but remain passive or "honest-but-curious," meaning they follow the learning protocol faithfully without modifying any update vectors (Truex et al., 2019; Peterson et al., 2019; Kerkouche et al., 2021). These adversaries can represent any party involved, such as a malicious client or server, aiming to extract information from other parties. The central server possesses knowledge of label classes for each client's data, while clients may or may not know the label classes of other clients' data. While we typically do not intentionally hide label class information among clients, our approach is flexible and can handle scenarios where clients want to keep their label class information confidential from others.

We integrate record-level DP into FedLAP to provide theoretical privacy guarantees, which yields FedLAP-DP. Given a desired privacy budget $(\varepsilon, \delta)$, we clip the gradients derived from real data with the Gaussian mechanism, denoted by $\nabla \widetilde{\mathcal{L}}(\mathbf{w}_k^{m,t}, \mathcal{D}_k)$. The DP-guaranteed local approximation can be realized by replacing the learning target of Eq. 9 with the gradients processed by DP while

leaving other constraints the same. Formally, we have

$$\underset{\mathcal{S}_k}{\arg\min} \sum_{t=1}^{T} \mathcal{L}_{\text{dis}}\Big(\nabla\widetilde{\mathcal{L}}(\mathbf{w}_k^{m,t}, \mathcal{D}_k), \nabla\mathcal{L}(\mathbf{w}_k^{m,t}, \mathcal{S}_k)\Big) \tag{12}$$
$$\text{s.t.} \quad \|\mathbf{w}_k^{m,t} - \mathbf{w}_k^{m,1}\| < r \quad \text{and} \quad \mathbf{w}_k^{m,t+1} = \mathbf{w}_k^{m,t} - \eta\nabla\mathcal{L}(\mathbf{w}_k^{m,t}, \mathcal{S}_k)$$

Note that $r$ is set to be a constant here to avoid additional privacy risks. We describe the full algorithm of FedLAP-DP in Algorithm 2 and present the privacy analysis of FedLAP-DP in the Sec. A. Our analysis suggests that with equivalent access to private data, FedLAP incurs the same privacy costs as gradient-sharing approaches. Our method further demonstrates a better privacy-utility trade-off in Sec. 5.3, confirming its robustness under DP noise.

## 5 EXPERIMENTS

### 5.1 SETUP

We consider a standard classification task by training federated ConvNets (LeCun et al., 2010) on three benchmark datasets: MNIST (LeCun et al., 1998), FashionMNIST (Xiao et al., 2017), and CIFAR-10 (Krizhevsky et al., 2009). Our study focuses on a non-IID setting where five clients possess disjoint class sets, meaning each client holds two unique classes. This scenario is typically considered challenging (Hsu et al., 2019) and mirrors the cross-silo setting (Kairouz et al., 2021) where all clients participate in every training round while maintaining a relatively large amount of data, yet exhibiting statistical divergence (e.g., envision the practical scenario for collaborations among hospitals). Our method employs a learning rate of 100 for updating synthetic images and 0.1 with cosine decay for model updates. We set by default $(R_i, R_l, R_b, r) = (4, 2, 10, 1.5)$ and $(1, 0, 5, 10)$ for DP and non-DP training, respectively. To prevent infinite loops caused by the neighborhood search, we upper bound the while loops in Algorithm 1 by 5 iterations. We follow FL benchmarks (McMahan et al., 2017; Reddi et al., 2021) and the official codes for training the baselines. All experiments are repeated over three random seeds. More details are provided in the Appendix.

### 5.2 DATA HETEROGENEITY

| | DSC[†] | FedSGD (1×) | FedAvg (1×) | FedProx (1×) | SCAFFOLD (2×) | FedDM (0.96×) | Ours (0.96×) |
|---|---|---|---|---|---|---|---|
| MNIST | 98.90±0.20 | 87.07±0.65 | 96.55±0.21 | 96.26±0.04 | 97.56±0.06 | 96.66±0.18 | **98.08±0.02** |
| Fa.MNIST | 83.60±0.40 | 75.10±0.16 | 79.67±0.56 | 79.37±0.29 | 82.17±0.37 | 83.10±0.16 | **87.37±0.09** |
| CIFAR-10 | 53.90±0.50 | 60.91±0.19 | **75.20±0.12** | 63.84±0.45 | 56.27±1.19 | 70.51±0.45 | 71.91±0.20 |

Table 1: Performance comparison on benchmark datasets. The relative communication cost of each method (w.r.t. the model size) is shown in brackets. DSC[†] is ported from the original paper and conducted in a one-shot centralized setting.

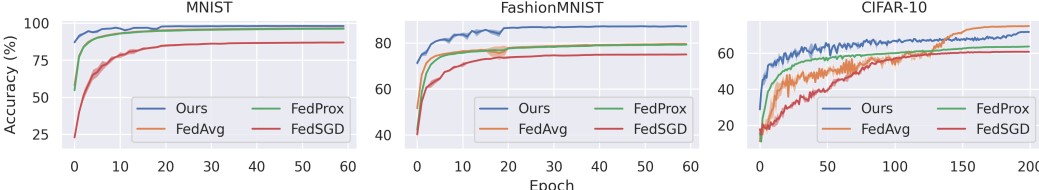

Figure 3: Accuracy over communication rounds with extremely non-IID data.

We first demonstrate the effectiveness of FedLAP over various baselines on benchmark datasets in a non-IID setting. Our method assigns 50 images to each class, resulting in comparable communication costs to the baselines. The baselines include: **DSC** (Zhao et al., 2021), the dataset distillation method considering centralized one-shot distillation; **FedSGD** (McMahan et al., 2017), that transmits every single batch gradient to prevent potential model drifting; **FedAvg** (McMahan et al., 2017), the most representative FL method; **FedProx** (Li et al., 2020), **SCAFFOLD** (Karimireddy et al., 2020), state-of-the-art federated optimization for non-IID distributions, and **FedDM** (Xiong

et al., 2023), a concurrent work that shares a similar idea but without considering approximation quality. Note that DSC operates in a (one-shot) centralized setting, SCAFFOLD incurs double the communication costs compared to the others by design, and FedDM requires class-wise optimization. As depicted in Table 1, our method surpasses DSC and FedSGD, highlighting the benefits of multi-round training on the server and client sides, respectively. Moreover, our method presents superior performance over state-of-the-art optimization methods, validating the strength of optimizing from a global view. We also plot model utility over training rounds in Fig. 3, where our method consistently exhibits the fastest convergence across three datasets. In other words, our methods consume fewer costs to achieve the same or better performance level and more communication efficiency.

## 5.3 PRIVACY PROTECTION

|  | PSG[†] Chen et al. (2022) | DP-FedAvg | DP-FedProx | Ours | DP-FedAvg | DP-FedProx | Ours |
|---|---|---|---|---|---|---|---|
| $\varepsilon$ | 32 | | 2.79 | | | 10.18 | |
| MNIST | 88.34±0.8 | 45.25±6.9 | 54.58±4.9 | **60.72±1.3** | 86.99±0.5 | **88.75±0.5** | 87.77±0.8 |
| FMNIST | 67.91±0.3 | 50.11±4.2 | 54.57±2.9 | **59.85±1.5** | 72.78±1.3 | 71.67±2.2 | **73.00±0.7** |
| CIFAR-10 | 34.58±0.4 | 17.11±0.7 | 19.40±0.7 | **21.42±1.4** | 31.15±0.4 | 35.04±1.1 | **36.09±0.5** |

Table 2: Utility and Privacy budgets at varying privacy regimes. The high privacy regime with $\varepsilon = 2.79$ corresponds to the first communication round, while a privacy level of $\varepsilon = 10.18$ represents the commonly considered point ($\varepsilon$=10) in private learning literature. PSG[†] corresponds to a one-shot centralized setting and is reproduced from the official code with the default configuration that yields $\varepsilon = 32$ in federated settings.

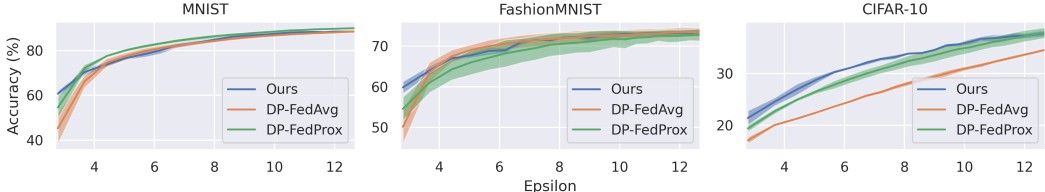

Figure 4: Privacy-utility trade-off with $\delta = 10^{-5}$. A *smaller* value of $\varepsilon$ (x-axis) indicates a *stronger* privacy guarantee. Evaluation is conducted at each communication round.

We evaluate the trade-off between utility and privacy costs $\varepsilon$ on benchmark datasets against two state-of-the-art methods, **DP-FedAvg** (the local DP version in Truex et al. (2019)) and **DP-FedProx**. Note that FedDM (Xiong et al., 2023) is incomparable since it considers class-wise optimization, introducing additional privacy risks and a distinct privacy notion. Our method assigns 10 images per class and is evaluated under the worst-case scenario, i.e., we assume the maximum of 5 while loops is always reached (Sec. 5.1) for the $\varepsilon$ computation, despite the potential early termination (and thus smaller $\varepsilon$) caused by the radius $r$ (Eq. 10). To ensure a fair and transparent comparison, we require our method to access the same amount of private data as the baselines in every communication round and consider a noise scale $\sigma = 1$ for all approaches. Fig. 4 demonstrates that our framework generally exhibits superior performance, notably with smaller $\varepsilon$ and more complex dataset such as CIFAR-10. This superiority is further quantified in Table 1 under two typical privacy budgets of 2.79 and 10.18. Moreover, when compared to the private one-shot dataset condensation method (**PSG** (Chen et al., 2022)), our approach presents a better privacy-utility trade-off, effectively leveraging the benefits of multi-round training in the challenging federated setting.

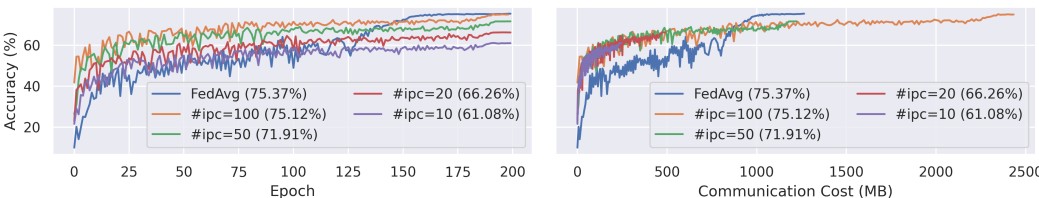

Figure 5: Ablation study on the number of images per class (#ipc).

## 5.4 ABLATION STUDY

**Radius Selection.**   As in Sec. 4, we assess various server optimization radius selection strategies: *Fixed*, *Max*, *Median*, and *Min*. The *Fixed* strategy employs a static length of 100 iterations regardless of quality, *Max* pursues swift optimization with the largest radius, *Median* moderates by adhering to the majority, and *Min*—used in all experiments—targets the safest region agreed by all synthetic image sets. Table 3 shows that strategies mindful of approximation quality surpass the fixed approach. Detailed analysis in Sec. D.2 reveals that aggressive strategies yield inferior intermediate performance, unsuitable for federated applications needing satisfactory intermediate results. Among the strategies, *Min* proves optimal.

**Size of synthetic datasets.**   We investigate the impact of synthetic dataset size on approximation. In general, higher numbers of synthetic samples submitted by clients lead to greater information communication. To further explore this concept, we conducted experiments on CIFAR-10, building upon the previous experiment (shown in Fig. 3) by adding three additional settings in which we assigned 10, 20, and 100 images to each class (referred to as "image per class" or #ipc). Our results, presented in Fig. 5, demonstrate that our method performs best when assigning 100 images, supporting the hypothesis that more synthetic samples convey more information. Additionally, our method produces superior outcomes regardless of #ipc when communication costs are restricted, making it advantageous for resource-constrained devices.

| Min | Max | Median | Fixed |
|---|---|---|---|
| 71.90 | 72.39 | 72.33 | 71.26 |

Table 3: Performance comparison between radius selection strategies.

## 6 DISCUSSION

**Future directions.**   While the primary contribution of FedLAP-DP lies in utilizing local approximation for global optimization, we demonstrate in the appendix that its performance can be further enhanced by improving the quality of the approximation. Moreover, ongoing research in synthetic data generation (Zhao et al., 2021; Zhao & Bilen, 2023; Cazenavette et al., 2022) represents a potential avenue for future work, which could potentially benefit our formulation.

**Computation overhead.**   Our method suggests an alternative to current research, trading computation for improved performance and communication costs incurred by slow convergence and biased optimization. We have empirically measured the computation time needed for a communication round by a client, using one NVIDIA Titan X, and observed an increase from 0.5 minutes (FedAvg) to 2.5 minutes (FedLAP). Despite this increase, the computation time is still manageable in cross-silo environments. A thorough analysis can be found in Section E. We anticipate this work will motivate the community to further explore the trade-off between computation and communication beyond local epochs, as we have shown in Fig. 5.

**(Visual) privacy.**   The initiative mentioned in (Dong et al., 2022) indicates that distilled data may offer enhanced privacy protection compared to plain gradients, and it is crucial to clarify that our synthetic images are *not crafted* to produce realistic or class-specific data. Nevertheless, a comprehensive privacy analysis concerning prevailing general dataset distillation is yet to be conducted and necessitates further examination across diverse scenarios.

## 7 CONCLUSION

In conclusion, this work introduces FedLAP-DP, a novel approach for privacy-preserving federated learning. FedLAP-DP utilizes synthetic data to approximate local loss landscapes within calibrated trust regions, effectively debiasing the optimization on the server. Moreover, our method seamlessly integrates record-level differential privacy, ensuring strict privacy protection for individual data records. Extensive experimental results demonstrate that FedLAP-DP outperforms gradient-sharing approaches in terms of faster convergence on highly-skewed data splits and reliable utility under differential privacy settings. We further explore the critical role of radius selection, the influence of synthetic dataset size, open directions, and potential enhancements to our work. Overall, FedLAP-DP presents a promising approach for privacy-preserving federated learning, addressing the challenges of convergence stability and privacy protection in non-IID scenarios.

## REPRODUCIBILITY STATEMENT

**Theoretical results.** In Sec. 3, we provide the background about federated learning, differential privacy, and theorems used in this work. We also provide detailed privacy analysis of the proposed FedLAP-DP in Sec. A, including the necessary definitions and notations, privacy composition for our iterative operations, and the conversion of our method from RDP to DP. Additional computation overhead is discussed in Sec. 6 and analyzed in Sec. E.

**Empirical results.** We provide the main hyper-parameters related to FedLAP-DP and the experiment settings in Sec. 5.1. The remaining details, including other hyper-parameters, architectures, learning rates, and hyper-parameter search, are provided in Sec. B. A detailed algorithm of FedLAP is presented in Algorithm 1 and the one of FedLAP-DP is in Algorithm 2. In addition to the hyper-parameter search in Sec. B, we analyze the necessity of magnitude regularization (Eq. 8) in Sec. D.1 and different radius strategies (Sec. 5.4 and Eq. 11) in Sec. D.2. The source code and setup will be anonymously provided as the forum opens and publicly available upon publication.

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

## A  PRIVACY ANALYSIS

### A.1  DEFINITIONS

**Definition A.1** (Rényi divergence). Let $P$ and $Q$ be two distributions defined over the same probability space $\mathcal{X}$. Let their respective densities be denoted as $p$ and $q$. The Rényi divergence, of a finite order $\alpha \neq 1$, between the distributions $P$ and $Q$ is defined as follows:

$$D_\alpha\left(P \parallel Q\right) \triangleq \frac{1}{\alpha - 1} \ln \int_{\mathcal{X}} q(x) \left(\frac{p(x)}{q(x)}\right)^\alpha dx \,.$$

Rényi divergence at orders $\alpha = 1, \infty$ are defined by continuity.

**Definition A.2** (Rényi differential privacy (RDP) Mironov (2017)). A randomized mechanism $\mathcal{M}:\mathcal{E} \to \mathcal{R}$ satisfies $(\alpha, \rho)$-Rényi differential privacy (RDP) if for any two adjacent inputs $E, E' \in \mathcal{E}$ it holds that

$$D_\alpha\left(\mathcal{M}(E) \parallel \mathcal{M}(E')\right) \leq \rho$$

In this work, we call two datasets $E, E'$ to be adjacent if $E' = E \cup \{\boldsymbol{x}\}$ (or vice versa).

**Definition A.3** (Sampled Gaussian Mechanism (SGM) Abadi et al. (2016); Mironov et al. (2019)). Let $f$ be an arbitrary function mapping subsets of $\mathcal{E}$ to $\mathbb{R}^d$. We define the Sampled Gaussian mechanism (SGM) parametrized with the sampling rate $0 < q \leq 1$ and the noise $\sigma > 0$ as

$$\mathrm{SG}_{q,\sigma} \overset{\Delta}{=} f\left(\{\boldsymbol{x} : \boldsymbol{x} \in E \text{ is sampled with probability } q\}\right) + \mathcal{N}(0, \sigma^2 \mathbb{I}_d),$$

where each element of $E$ is independently and randomly sampled with probability $q$ without replacement.

The sampled Gaussian mechanism consists of adding i.i.d Gaussian noise with zero mean and variance $\sigma^2$ to each coordinate of the true output of $f$. In fact, the sampled Gaussian mechanism draws random vector values from a multivariate isotropic Gaussian distribution denoted by $\mathcal{N}(0, \sigma^2 \mathbb{I}_d)$, where $d$ is omitted if it is unambiguous in the given context.

## A.2 ANALYSIS

The privacy analysis of FedLAP-DP and other DP baselines follows the analysis framework used for gradient-based record level-DP methods in FL Truex et al. (2019); Peterson et al. (2019); Kerkouche et al. (2021). In this framework, each individual local update is performed as a single SGM (Definition A.3) that involves clipping the per-example gradients on a local batch and subsequently adding Gaussian noise to the averaged batch gradient (Algorithm 2). The privacy cost accumulated over multiple local updates and global rounds is then quantified utilizing the revisited moment accountant Mironov et al. (2019), which presents an adapted version of the moments accountant introduced in Abadi et al. (2016) by adapting to the notion of RDP (Definition A.2). Finally, to obtain interpretable results and enable transparent comparisons to established approaches, we convert the privacy cost from $(\alpha, \rho)$-RDP to $(\varepsilon, \delta)$-DP by employing Theorem A.7 provided by Balle et al. (2020).

**Theorem A.4.** *Mironov et al. (2019) Let* $\mathrm{SG}_{q,\sigma}$ *be the Sampled Gaussian mechanism for some function $f$ with $\Delta_2 f \leq 1$ for any adjacent $E, E' \in \mathcal{E}$. Then $\mathrm{SG}_{q,\sigma}$ satisfies $(\alpha, \rho)$-RDP if*

$$\rho \leq D_\alpha\left(\mathcal{N}(0, \sigma^2) \,\big\|\, (1-q)\mathcal{N}(0, \sigma^2) + q\mathcal{N}(1, \sigma^2)\right)$$

$$and \quad \rho \leq D_\alpha\left((1-q)\mathcal{N}(0, \sigma^2) + q\mathcal{N}(1, \sigma^2) \,\big\|\, \mathcal{N}(0, \sigma^2)\right)$$

Theorem A.4 reduce the problem of proving the RDP bound for $\mathrm{SG}_{q,\sigma}$ to a simple special case of a mixture of one-dimensional Gaussians.

**Theorem A.5.** *Mironov et al. (2019). Let $\mu_0$ denote the pdf of $\mathcal{N}(0, \sigma^2)$, $\mu_1$ denote the pdf of $\mathcal{N}(1, \sigma^2)$, and let $\mu$ be the mixture of two Gaussians $\mu = (1-q)\mu_0 + q\mu_1$. Let $\mathrm{SG}_{q,\sigma}$ be the Sampled Gaussian mechanism for some function $f$ and under the assumption $\Delta_2 f \leq 1$ for any adjacent $E, E' \in \mathcal{E}$. Then $\mathrm{SG}_{q,\sigma}$ satisfies $(\alpha, \rho)$-RDP if*

$$\rho \leq \frac{1}{\alpha - 1} \log\left(\max\{A_\alpha, B_\alpha\}\right) \tag{13}$$

*where $A_\alpha \overset{\Delta}{=} \mathbb{E}_{z \sim \mu_0}[(\mu(z)/\mu_0(z))^\alpha]$ and $B_\alpha \overset{\Delta}{=} \mathbb{E}_{z \sim \mu}[(\mu_0(z)/\mu(z))^\alpha]$*

Theorem A.5 states that applying SGM to a function of sensitivity (Eq. 5) at most 1 (which also holds for larger values without loss of generality) satisfies $(\alpha, \rho)$-RDP if $\rho \leq \frac{1}{\alpha-1} \log(\max\{A_\alpha, B_\alpha\})$. Thus, analyzing RDP properties of SGM is equivalent to upper bounding $A_\alpha$ and $B_\alpha$.

From Corollary 7 in Mironov et al. (2019), $A_\alpha \geq B_\alpha$ for any $\alpha \geq 1$. Therefore, we can reformulate 13 as

$$\rho \leq \xi_{\mathcal{N}}(\alpha|q) := \frac{1}{\alpha - 1} \log A_\alpha \qquad (14)$$

To compute $A_\alpha$, we use the numerically stable computation approach proposed in Section 3.3 of Mironov et al. (2019). The specific approach used depends on whether $\alpha$ is expressed as an integer or a real value.

**Theorem A.6** (Composability Mironov (2017)). *Suppose that a mechanism $\mathcal{M}$ consists of a sequence of adaptive mechanisms $\mathcal{M}_1, \ldots, \mathcal{M}_k$ where $\mathcal{M}_i : \prod_{j=1}^{i-1} \mathcal{R}_j \times \mathcal{E} \to \mathcal{R}_i$. If all mechanisms in the sequence are $(\alpha, \rho)$-RDP, then the composition of the sequence is $(\alpha, k\rho)$-RDP.*

In particular, Theorem A.6 holds when the mechanisms themselves are chosen based on the (public) output of the previous mechanisms. By Theorem A.6, it suffices to compute $\xi_{\mathcal{N}}(\alpha|q)$ at each step and sum them up to bound the overall RDP privacy budget of an iterative mechanism composed of single DP mechanisms at each step.

**Theorem A.7** (Conversion from RDP to DP Balle et al. (2020)). *If a mechanism $\mathcal{M}$ is $(\alpha, \rho)$-RDP then it is $((\rho + \log((\alpha - 1)/\alpha) - (\log \delta + \log \alpha)/(\alpha - 1), \delta)$-DP for any $0 < \delta < 1$.*

**Theorem A.8** (Privacy of FedLAP-DP). *For any $0 < \delta < 1$ and $\alpha \geq 1$, FedLAP-DP is $(\varepsilon, \delta)$-DP, with*

$$\varepsilon = \min_{\alpha} \left( M \cdot \xi_{\mathcal{N}}(\alpha|q_1) + M(T-1) \cdot \xi_{\mathcal{N}}(\alpha|q_2) + \log((\alpha-1)/\alpha) - (\log \delta + \log \alpha)/(\alpha - 1) \right) \quad (15)$$

*Here, $\xi_{\mathcal{N}}(\alpha|q)$ is defined in Eq. 14, $q_1 = \frac{C \cdot \mathbb{B}}{\min_k |\mathcal{D}_k|}$, $q_2 = \frac{\mathbb{B}}{\min_k |\mathcal{D}_k|}$, $M$ is the number of federated rounds, $T$ is the total number of local updates (i.e., total accesses to local private data) per federated round, $C$ is the probability of selecting any client per federated round, $\mathbb{B}$ is the local batch size, and $|\mathcal{D}_k|$ denotes the local dataset size.*

The proof follows from Theorems A.5, A.6, A.7 and the fact that a record is sampled in the very first SGD iteration of every round if two conditions are met. First, the corresponding client must be selected, which occurs with a probability of $C$. Second, the locally sampled batch at that client must contain the record, which has a probability of at most $\frac{\mathbb{B}}{\min_k |\mathcal{D}_k|}$. However, the adaptive composition of consecutive SGD iterations are considered where the output of a single iteration depends on the output of the previous iterations. Therefore, the sampling probability for the first batch is $q_1 = \frac{C \cdot \mathbb{B}}{\min_k |\mathcal{D}_k|}$, while the sampling probability for every subsequent SGD iteration within the same round is at most $q_2 = \frac{\mathbb{B}}{\min_k |\mathcal{D}_k|}$ *conditioned* on the result of the first iteration Kerkouche et al. (2021).

## B SETUP DETAILS

We consider a standard classification task by training federated ConvNets LeCun et al. (2010) on three benchmark datasets: MNIST LeCun et al. (1998), FashionMNIST Xiao et al. (2017), and CIFAR-10 Krizhevsky et al. (2009). Our study focuses on the non-IID setting where five clients possess disjoint class sets, meaning each client holds two unique classes. This scenario is typically considered challenging Hsu et al. (2019) and mirrors the cross-silo setting Kairouz et al. (2021) where all clients participate in every training round while maintaining a relatively large amount of data, yet exhibiting statistical divergence (e.g., envision the practical scenario for collaborations among hospitals).

By default, we set the hyperparameters $(R_i, R_l, R_b, r)$ to be $(4, 2, 10, 1.5)$ and $(1, 0, 5, 10, 0.1)$ for DP (Differential Privacy) and non-DP training, respectively. Additionally, we include a weight of $0.1$ for Mean Squared Error (MSE) regularization in our method (Eq. 9 and 16. To avoid infinite loops caused by neighborhood search, we limit the iterations in Algorithm 1 (referred to as Algorithm 1 in the original text) to a maximum of 5.

Our method utilizes a learning rate of 100 for updating synthetic images, and we specify the learning schedulers as described below. For training the baselines, we adhere to the FL (Federated Learning) benchmarks McMahan et al. (2017); Reddi et al. (2021) and use the official codes. All experiments

are conducted on a single Titan RTX GPU and repeated three times with different random seeds. Further implementation details are provided in the following sections. The source code will be made publicly available upon publication.

## B.1 ARCHITECTURES

We provide the details of the federated ConvNet used in our paper. The network consists of three convolutional layers, followed by two fully-connected layers. ReLU activation functions are applied between each layer. Each convolution layer, except for the input layer, is composed of 128 (input channels) and 128 (output channels) with $3 \times 3$ filters. Following prior work Wang et al. (2022); Reddi et al. (2021); Zhao & Bilen (2023); Cazenavette et al. (2022), we attach Group Normalization Wu & He (2018) before the activation functions to stabilize training. For classification, the network utilizes a global average pooling layer to extract features, which are then fed into the final classification layer for prediction. The entire network contains a total of 317,706 floating-point parameters. The model details are listed below.

```
ConvNet(
  (features): Sequential(
    (0): Conv2d(3, 128, kernel_size=(3, 3), stride=(1, 1), padding=(1, 1))
    (1): GroupNorm(128, 128, eps=1e-05, affine=True)
    (2): ReLU(inplace=True)
    (3): AvgPool2d(kernel_size=2, stride=2, padding=0)
    (4): Conv2d(128, 128, kernel_size=(3, 3), stride=(1, 1), padding=(1, 1))
    (5): GroupNorm(128, 128, eps=1e-05, affine=True)
    (6): ReLU(inplace=True)
    (7): AvgPool2d(kernel_size=2, stride=2, padding=0)
    (8): Conv2d(128, 128, kernel_size=(3, 3), stride=(1, 1), padding=(1, 1))
    (9): GroupNorm(128, 128, eps=1e-05, affine=True)
    (10): ReLU(inplace=True)
    (11): AvgPool2d(kernel_size=2, stride=2, padding=0)
  )
  (classifier): Linear(in_features=2048, out_features=10, bias=True)
)
```

## B.2 NON-DP

We maintain a consistent total of 60 communication rounds for MNIST and FashionMNIST, and 200 rounds for CIFAR-10. Following prior work McMahan et al. (2017); Reddi et al. (2021), we tune the hyper-parameters for various gradient-sharing schemes, including FedSGD McMahan et al. (2017), FedAvg McMahan et al. (2017), FedProx Li et al. (2020), and SCAFFOLD Karimireddy et al. (2020). We use a client learning rate of 0.01 for MNIST and FashionMNIST and 0.1 for CIFAR-10. All baselines are equipped with a fixed server learning rate of 1 and a cosine learning rate decay for the clients (potentially diverge, if without decay Reddi et al. (2021)). While hyper-parameter searching is an active field and requires non-trivial effort Khodak et al. (2021), we explore a proper number of local epochs by running FedAvg for 200 rounds with $\{1, 3, 5\}$ local epochs, leading to 69.13, 69.39, and 71.91, respectively. This suggests that 5 local epochs work best for FedAvg. We also set 5 local epochs for the rest of baselines for a fair comparison regarding data exploitation and communication efficiency. For FedProx, we assign a weight of 0.1 for the proximal regularization term.

In the case of FedLAP-DP, we use a server learning rate of 0.01 for MNIST and FashionMNIST, and 0.1 for CIFAR-10. These learning rates match the client learning rates used in the gradient-sharing baselines, while shifting the potentially biased local optimization to the server side. Additionally, our method adopts the same cosine learning rate decay as the baselines. These settings ensure a fair comparison as the parameters are not biased in favor of our method in non-private settings. Furthermore, in each communication round, we assign 50 images to each class and re-initialize the synthetic images from scratch. After receiving the images, the server performs full gradient descent to update the model parameters. This process ensures that the model is updated based on the collective knowledge obtained from the synthetic images generated by the clients.

|  | DP-FedAvg (low) | DP-FedAvg (high) | Ours | DP-FedAvg (low) | DP-FedAvg (high) | Ours |
|---|---|---|---|---|---|---|
| $\varepsilon$ | | 2.79 | | | 10.18 | |
| MNIST | 16.25 | 45.25 | **60.72** | 51.53 | 86.99 | **87.77** |
| FashionMNIST | 13.39 | 50.11 | **59.85** | 59.26 | 72.78 | **73.00** |
| CIFAR-10 | 9.37 | 17.11 | **21.42** | 16.16 | 31.15 | **36.09** |

Table 4: Comparison of DP-FedAvg with different learning rates: 0.1 (DP-FedAvg high) and 0.002 (DP-FedAvg low). We use the same initial learning rate as our method (0.002) for DP-FedAvg (low).

### B.3 DP

In this experiment, we compare the performance of FedLAP-DP to DP-FedAvg and DP-FedProx. We set a learning rate of 0.1 with cosine decay for the baselines. Following the approach of Kurakin et al. (2022), we initially verify that the learning rate works well in non-private settings. Then, we search for the sensitivity value from a grid of 1.0, 0.5, 0.2, 0.1. This results in a sensitivity of 0.1 for DP-FedAvg, 0.2 for DP-FedProx, and 1.0 for FedLAP-DP. Moreover, our method assigns 10 images to each class to mitigate the negative impact of differential privacy (DP) noise and a lower learning rate of 0.002, coupled with cosine decay, for all DP experiments.

## C    ALGORITHMS

**Distance metric**    Following existing works Zhao et al. (2021), the distance $\mathcal{L}_{\mathrm{dis}}$ (in Equation 8 and 9 of the main paper) between the real and synthetic gradients is defined to be the sum of the cosine distance at each layer. Let $\mathbf{w}^{(l)}$ denote the weight at the $l$-th layer, the distance can be formularized as:

$$\mathcal{L}_{\mathrm{dis}}\Big(\nabla_{\mathbf{w}}\mathcal{L}(\mathbf{w}, \mathcal{D}_k), \nabla_{\mathbf{w}}\mathcal{L}(\mathbf{w}, \mathcal{S}_k)\Big)$$

$$= \sum_{l=1}^{L} d\Big(\nabla_{\mathbf{w}^{(l)}}\mathcal{L}(\mathbf{w}^{(l)}, \mathcal{D}_k), \nabla_{\mathbf{w}^{(l)}}\mathcal{L}(\mathbf{w}^{(l)}, \mathcal{S}_k)\Big) + \lambda\|\nabla_{\mathbf{w}^{(l)}}\mathcal{L}(\mathbf{w}^{(l)}, \mathcal{D}_k) - \nabla_{\mathbf{w}^{(l)}}\mathcal{L}(\mathbf{w}^{(l)}, \mathcal{S}_k)\|_2^2$$

$$(16)$$

$d$ denotes the cosine distance between the gradients at each layer:

$$d(\boldsymbol{A}, \boldsymbol{B}) = \sum_{i=1}^{out} \left(1 - \frac{\boldsymbol{A}_{i\cdot} \cdot \boldsymbol{B}_{i\cdot}}{\|\boldsymbol{A}_{i\cdot}\|\|\boldsymbol{B}_{i\cdot}\|}\right) \tag{17}$$

where $\boldsymbol{A}_{i\cdot}$ and $\boldsymbol{B}_{i\cdot}$ represent the flattened gradient vectors corresponding to each output node $i$. In FC layers, $\boldsymbol{\theta}^l$ is a 2D tensor with dimension $out \times in$ and the flattened gradient vector has dimension $in$, while in Conv layers, $\boldsymbol{\theta}^l$ is a 4D tensor with dimensionality $out \times in \times h \times w$ and the flattened vector has dimension $in \times h \times w$. Here we use $out$, $in$, $h$, $w$ to denote the number of output and input channels, kernel height, and width, respectively.

**FedLAP-DP**    We present the pseudocode of the FedLAP-DP with record-level DP in Algorithm 2, which is supplementary to Sec. 4.4 of the main paper.

---

**Algorithm 2** FedLAP-DP

---

**function** ServerExecute:
    **Initialize** global weight $\mathbf{w}_g^{1,1}$, Fix the radius $r$
    /* Local approximation */
    **for** $m = 1, \ldots, M$ **do**
        **for** $k = 1, \ldots, K$ **do**
            $\mathcal{S}_k \leftarrow$ ClientsExecute($k, r, \mathbf{w}_g^{m,1}$)
        **end for**
        /* Global optimization */
        $t \leftarrow 1$
        **while** $\|\mathbf{w}_g^{m,1} - \mathbf{w}_g^{m,t}\| < r$ **do**
            $\mathbf{w}_g^{m,t+1} = \mathbf{w}_g^{m,t} - \sum_{k=1}^{K} \eta \frac{1}{K} \nabla_{\mathbf{w}} \mathcal{L}(\mathbf{w}_g^{m,t}, \mathcal{S}_k)$
            $t \leftarrow t + 1$
        **end while**
        $\mathbf{w}_g^{m+1,1} \leftarrow \mathbf{w}_g^{m,t}$
    **end for**
**Return:** global model weight $\mathbf{w}_g^{M+1,1}$

---

**function** ClientExecute($k, r, \mathbf{w}_g^{m,1}$) :
    **Initialize** $\mathcal{S}_k$: $\{\hat{\boldsymbol{x}}_k^m\}$ from Gaussian noise or $\{\hat{\boldsymbol{x}}_k^{m-1}\}$, $\{\hat{y}_k\}$ to be a balanced set
    **for** $i = 1, \ldots, R_i$ **do**
        /* Resample training trajectories */
        Reset $t \leftarrow 1$ and model $\mathbf{w}_k^{m,1} \leftarrow \mathbf{w}_g^{m,1}$
        **while** $\|\mathbf{w}_k^{m,t} - \mathbf{w}_k^{m,1}\| < r$ **do**
            Uniformly sample random batch $\{(\boldsymbol{x}_k^i, y_k^i)\}_{i=1}^{\mathbb{B}}$ from $\mathcal{D}_k$
            **for** each $(\boldsymbol{x}_k^i, y_k^i)$ **do**
                /* Compute per-example gradients on client data */
                $g^{\mathcal{D}}(\boldsymbol{x}_k^i) = \nabla_{\mathbf{w}} \ell(\mathbf{w}_k^{m,t}, \boldsymbol{x}_k^i, y_k^i)$
                /* Clip gradients with bound $\mathbb{C}$ */
                $\widetilde{g^{\mathcal{D}}}(\boldsymbol{x}_k^i) = g^{\mathcal{D}}(\boldsymbol{x}_k^i) \cdot \min(1, \mathbb{C}/\|g^{\mathcal{D}}(\boldsymbol{x}_k^i)\|_2)$
            **end for**
            /* Add noise to average gradient by Gaussian mechanism */
            Compute $\nabla \widetilde{\mathcal{L}}(\mathbf{w}_k^{m,t}, \mathcal{D}_k) = \frac{1}{\mathbb{B}} \sum_{i=1}^{\mathbb{B}} (\widetilde{g^{\mathcal{D}}}(\boldsymbol{x}_k^i) + \mathcal{N}(0, \sigma^2 \mathbb{C}^2 I))$
            **for** $j = 1, \ldots, R_b$ **do**
                /* Update synthetic set $\mathcal{S}_k$ */
                $\mathcal{S}_k = \mathcal{S}_k - \tau \nabla_{\mathcal{S}_k} \mathcal{L}_{dis}\left(\nabla \widetilde{\mathcal{L}}(\mathbf{w}_k^{m,t}, \mathcal{D}_k), \nabla \mathcal{L}(\mathbf{w}_k^{m,t}, \mathcal{S}_k)\right)$
            **end for**
            **for** $l = 1, \ldots, R_l$ **do**
                /* Update local model parameter $\mathbf{w}_k$ */
                $\mathbf{w}_k^{m,t+1} = \mathbf{w}_k^{m,t} - \eta \nabla \mathcal{L}(\mathbf{w}_k^{m,t}, \mathcal{S}_k)$
                $t \leftarrow t + 1$
            **end for**
        **end while**
    **end for**
**Return:** Synthetic set $\mathcal{S}_k$

---

# D ADDITIONAL ANALYSIS

## D.1 MATCHING CRITERIA

We analyze our design choices by using our method to fit gradients computed on a single batch. We simplify the learning task by employing only one client that contains all training data, resembling centralized training or FedSGD with a single client. In this setup, the client immediately communicates the gradients to the central server after computing them on a single batch. This scenario can

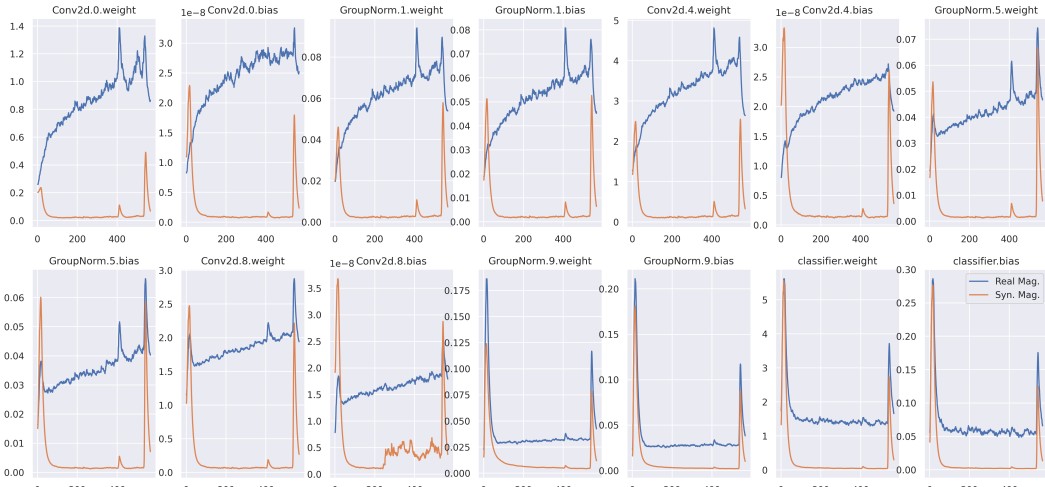

Figure 6: Discrepancy in gradient magnitudes between real and synthetic data. The noticeable difference in magnitudes during training highlights the limitation of solely regulating gradient directions, as optimizing without magnitude information introduces training instability.

be seen as a trivial federated learning task, as it does not involve any model drifting (non-IID) or communication budget constraints. It is worth noting that the baseline performance in this setting is an *ideal* case that does not apply in any practical FL use cases (or out of scope of federated learning).

**Gradient Magnitudes** While previous research Zhao et al. (2021); Zhao & Bilen (2021); He et al. (2021) suggests that gradient directions are more crucial than magnitudes (Eq. 17), our study demonstrates that as training progresses, the magnitudes of synthetic gradients (i.e., gradients obtained from synthetic images) can differ significantly from real gradients. In Fig. 6, we display the gradient magnitudes of each layer in a ConvNet. Our findings indicate that even with only 500 iterations, the magnitudes of synthetic gradients (orange) noticeably deviate from the real ones (blue), causing unnecessary instability during training.

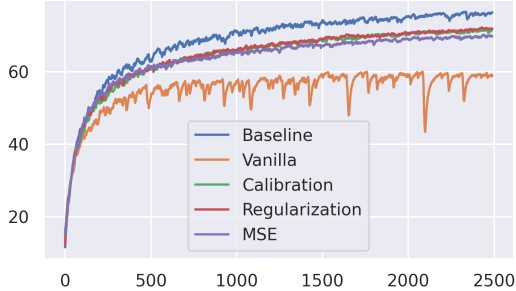

Figure 7: Performance comparison of fitting one batch. **Baseline**: FedSGD with one client. **Vanilla**: our method with cosine similarity. **Calibration**: our method with cosine similarity and magnitude calibration. **Regularization**: our method with cosine similarity and MSE regularization (i.e., Eq. 16). **MSE**: gradient matching by solely measuring mean square errors (Eq. 19).

**Post-hoc Magnitude Calibration** To further validate the issue, we implement a post-hoc magnitude calibration, called *Calibration* in Fig. 7. It calibrates the gradients obtained from the synthetic images on the server. Specifically, the clients send the layer-wise magnitudes of real gradients $\|\nabla_{\mathbf{w}}\mathcal{L}(\mathbf{w},\mathcal{D}_k)\|$ to the server, followed by a transformation on the server:

$$\frac{\nabla_{\mathbf{w}}\mathcal{L}(\mathbf{w},\mathcal{S}_k)}{\|\nabla_{\mathbf{w}}\mathcal{L}(\mathbf{w},\mathcal{S}_k)\|}\|\nabla_{\mathbf{w}}\mathcal{L}(\mathbf{w},\mathcal{D}_k)\|. \tag{18}$$

In Fig. 7, We observe that the synthetic images with the magnitude calibration successfully and continuously improve over the one without the calibration (Vanilla). It implies that even with the same cosine similarity, inaccurate gradient magnitudes could dramatically fail the training. It is worth noting that the performance gap between the baseline and our method in this experiment does not apply to federated learning since FL models suffer from non-IID problems induced by multiple steps and clients, the gradient-sharing schemes, such as FedAvg, overly approximate the update

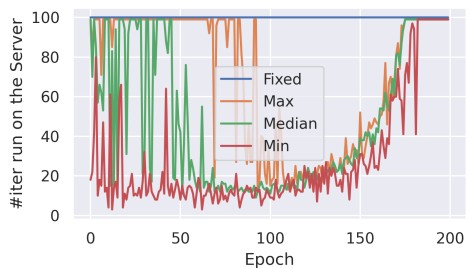

Figure 8: Training iterations for different radius selection strategies.

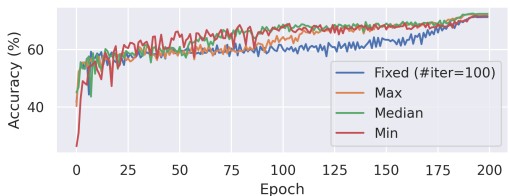

Figure 9: Ablation study on radius selection.

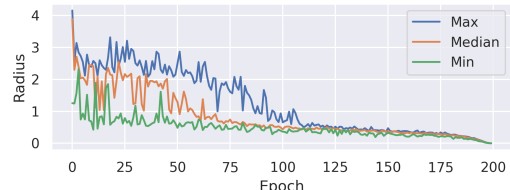

Figure 10: Radius suggested by different strategies.

signal, further enhancing the problem. Meanwhile, it also suggests an opportunity to improve our method if future work can further bridge the gap.

**Mean Square Error (MSE) Regularization**  Although calibration can improve performance, it is not suitable for federated learning due to two reasons. Firstly, when the synthetic images $\mathcal{S}_k$ from different clients are merged into a dataset for server optimization, it remains unclear how to apply Eq. 18 to the averaged gradients of the merged dataset $\mathcal{S}$, especially when multiple-step optimization is involved. Secondly, transmitting magnitudes can pose privacy risks. Instead of explicit calibration, we propose using MSE regularization (Eq. 16, termed *Regularization* in Fig. 7) to limit potential magnitude deviations while focusing on the directions. As depicted in Fig. 7, our proposed method remains close to the calibration method, suggesting that regularization prevents mismatch.

Moreover, we present an additional implementation with an MSE matching criterion (termed *MSE* in Fig. 7)). It solely matches the mean square error distance between $\mathcal{L}(\mathbf{w}, \mathcal{D}_k)$ and $\mathcal{L}(\mathbf{w}, \mathcal{S}_k)$ regardless of gradient directional information. That is,

$$\|\nabla_{\mathbf{w}^{(l)}} \mathcal{L}(\mathbf{w}^{(l)}, \mathcal{D}_k) - \nabla_{\mathbf{w}^{(l)}} \mathcal{L}(\mathbf{w}^{(l)}, \mathcal{S}_k)\|_2^2 \tag{19}$$

Despite the improvement over the vanilla method, it still falls behind *Calibration* and *Regularization*, highlighting the importance of directional information and justifying our design choice.

### D.2    Radius Selection

As in Sec. 4, we evaluate different radius selection strategies for server optimization. We consider four strategies: *Fixed*, *Max*, *Median*, and *Min*. *Fixed* uses a fixed length of 100 iterations, ignoring the quality. *Max* aims for the fastest optimization by using the largest radius. *Median* optimizes in a moderate way by considering the majority. *Min*, adopted in

| Min | Max | Median | Fixed |
|-----|-----|--------|-------|
| 71.90 | 72.39 | 72.33 | 71.26 |

Table 5: Performance comparison between radius selection strategies.

all experiments, focuses on the safest region agreed upon by all synthetic image sets. Table 3 reveals that the proposed strategies consistently outperform the fixed strategy. Meanwhile, Fig. 9 presents that a more aggressive strategy leads to worse intermediate performance, which may not be suitable for federated applications requiring satisfactory intermediate performance. Among them, *Min* delivers the best results. Finally, Fig. 10 demonstrates that the radii proposed by different strategies

change across epochs, indicating that a naively set training iteration may not be optimal. Additionally, this finding suggests the possibility of designing heuristic scheduling functions for adjusting the radius in a privacy-preserving way. The corresponding server training iterations can be found in the appendix.

In Sec. 5.4, we show that the effective approximation regions change across rounds. A fixed predefined training iterations may cause sub-optimal performance. To complement the experiments, we additionally plot the corresponding training iterations on the server side in Fig. 8. We observed that *Max* and *Median* tend to be more aggressive by updating for more epochs, granting faster improvement, while *Min* optimizes more conservatively. Interestingly, we found that all three proposed strategies exhibit similar behavior in the later stages of training, which is in stark contrast to the fixed strategy.

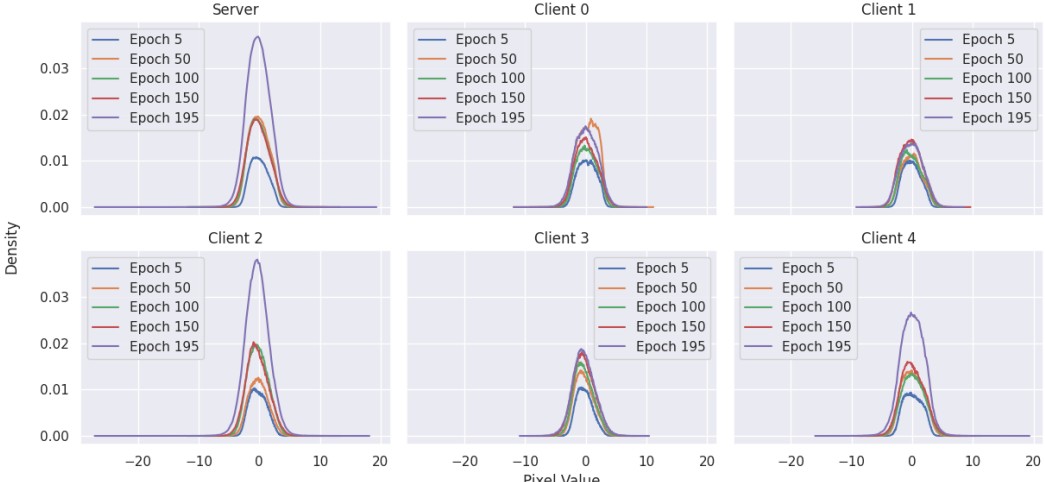

Figure 11: Pixel distributions in non-private settings. The plot illustrates the evolution of pixel values in synthetic images during training. At the early training stage, the pixel range is wider and gradually concentrates around zero as the model approaches convergence, implying that the synthetic images reflects the training status.

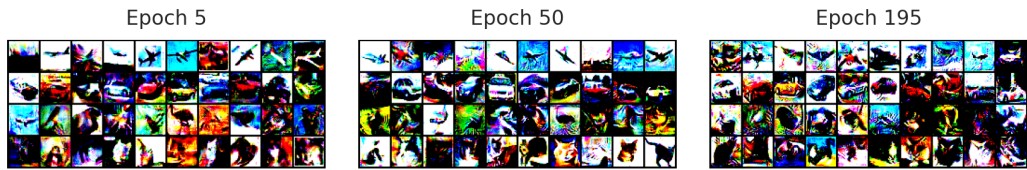

Figure 12: Visualization of synthetic images at epochs 5, 50, and 196 in the non-private CIFAR-10 experiment. The pixel values are clipped to the range $[0, 1]$. Each rows corresponds to airplane, automobile, bird, and cat, respectively.

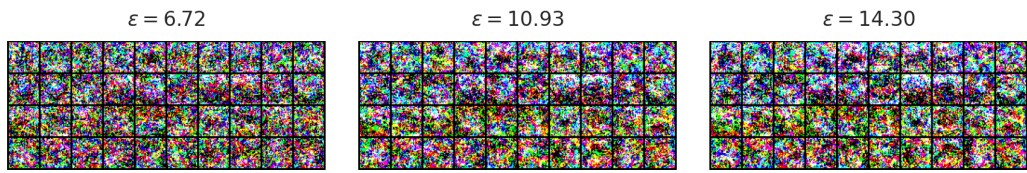

Figure 13: Visualization of synthetic images for $\varepsilon = 6.72, 10.93$, and $14.30$ in the privacy-preserving CIFAR-10 experiment. The pixel values are clipped to the range $[0, 1]$. Each rows corresponds to airplane, automobile, bird, and cat, respectively.

### D.3 QUALITATIVE RESULTS

In this section, we examine the synthetic images. In non-private settings, Fig. 11 displays the pixel value distributions of all synthetic images at epochs 5, 50, 100, 150, and 195, both on the server and the clients. We made several observations from these visualizations. Firstly, within the same round, the distributions of pixel values on each client exhibit distinct behavior, indicating the diversity of private data statistics across clients. Secondly, the distributions also vary across different epochs. At earlier stages, the synthetic images present a wider range of values, gradually concentrating around zero as training progresses. This phenomenon introduces more detailed information for training and demonstrates that our method faithfully reflects the training status of each client. Additionally, Fig. 12 displays visual examples of synthetic images corresponding to the labels "airplane," "automobile," "bird," and "cat." These visuals indicate that earlier epoch images exhibit larger pixel values and progressively integrate more noise during training, reflecting gradient details. It's crucial to underscore that our method is *not intended* to produce realistic data but to approximate loss landscapes.

## E COMPUTATION COMPLEXITY

Our method suggests an alternative to current research, trading computation for improved performance and communication costs incurred by slow convergence and biased optimization. We present the computation complexity analysis for one communication round below and conclude with empirical evidence that the additional overhead is manageable, especially for cross-silo scenarios.

We begin with the SGD complexity $\mathcal{O}(d)$, where $d$ denotes the number of network parameters. Suppose synthetic samples contain $p$ trainable parameters; then the complexity can be formulated as follows.

$$\mathcal{O}\bigg( R_i \cdot 5 \cdot \Big( N(d + 2R_b(d + p)) + R_l d \Big) \bigg)$$
$$= \mathcal{O}\bigg( 5R_i \cdot (2NR_b + R_l + N)d + 10R_iR_bp \bigg)$$
$$= \mathcal{O}\bigg( 5R_iN \cdot (2R_b + \frac{R_l}{N} + 1)d + 10R_iR_bp \bigg)$$

Note that $5R_iN$ determines how much real data we will see during synthesis. For comparison, we make $5R_iN$ equal in both our method and gradient-sharing baselines (i.e., five local epochs in FedAvg with complexity $\mathcal{O}(5R_iNd)$). Overall, our method introduces $2R_b + \frac{R_l}{N} + 1$ times more computation on network parameters and an additional $10R_iR_b$ term for updating synthetic samples.

