# OpenReview forum: "FedLAP-DP: Federated Learning by Sharing Differentially Private Loss Approximations"
_ICLR.cc/2024/Conference — ICLR 2024 Conference Withdrawn Submission_

### Official Review · Reviewer_vGpK · 2023-10-25

**Soundness:** 2 fair
**Presentation:** 2 fair
**Contribution:** 3 good
**Rating:** 5
**Confidence:** 3

**Summary:**

This paper proposes a new federated learning training algorithm. The key idea of the algorithms is to share synthetic datasets instead of sharing the model weights. The authors argue that the method is superior to traditional model-averaging because the server can recover an approximation of the global loss landscape. The authors also demonstrate that the proposed method can be adapted to satisfy differential privacy (DP) by replacing the clean gradients with clipped noisy gradients. Some experiments show that the new method can outperform the traditional one in both private and non-private settings and may have better communication cost tradeoffs.

**Strengths:**

* The authors identify the problem of the existing weight-averaging FL training methods.
* The proposed method can overcome the limitations of the existing algorithms.
* The proposed algorithm is shown to have better empirical performances than the existing ones.

**Weaknesses:**

1. Some operations in the proposed algorithm may need better motivation or explanation. Please refer to Questions.
2. The privacy description part of the algorithm can be improved. The description in Section 4.4 is not clear enough and self-contained for a main contribution in the main text. Some descriptions may need to be more precise (e.g., "sanitize" should be explicitly referred to the clipping operation). Since the privacy composition part mainly relies on existing tools, it may be better to provide a brief conclusion about the composition results. Also, the notations in Appendix (T?) are not consistent with Algorithm 1 (R_b?), which requires extra conjecture to parse the result.
3. Some hyper-parameters may need to show how to tune ($R_b$ and $R_l$), because they are so different for private and non-private settings.
4. The experiment setting may need to be more convincing. Please refer to Questions.

**Questions:**

1. It is mentioned that the synthetic labels are initialized to be a fixed, balanced set, but the experiments have heterogeneous data. This sounds controversial and may deserve more explanation. When a class does not appear in a local dataset, what should we expect the synthetic data of that class to look like? How do those affect global training compared with relatively iid data?
2. How do you decide the synthetic dataset size of each client? Besides the factors mentioned (local dataset size and bandwidth), it seems the data complexity and local data distribution should also be considered when deciding the synthetic dataset size.
3. Section 5.1 mentions that the learning rate is 100. Is it a typo? Otherwise, why we need such a large learning rate may need to be explained.
4. What do the "Fixed, Max, Median, and Min" radius selection mean? Do they matter for whether private or non-private settings? The description needs to be more specific and consistent (i.e., it says r=1.5 or 10 in Section 5.1, but a different wording in Section 5.4).
5. In the experiment setting of comparing the communication cost and epochs, there may be more reasonable settings. For example, when comparing the communication cost, shouldn't we compare the best end-to-end performance with fixed communication costs (e.g., 500MB) by varying the communication rounds and with the best hyper-parameters?
6. Another interesting aspect not explored in the experiments FedAvg v.s. FedLAP with different sizes of models. Does a model with more parameters benefit from or loss advantage with the proposed method?

---

> ### Author Response · Authors · 2023-11-16
> **Response to Reviewer vGpK**
>
> We sincerely thank the reviewer for the insightful comments and will polish the paper accordingly. We now address the specific questions and concerns below.
>
> ----
> **Q1: How to tune ($R_b$ and $R_l$)**
>
> The parameter $R_b$ is not contingent on privacy budgets and generally enhances performance when assigned a higher value. We chose different values to limit computation and execution time. Regarding $R_l$, we conduct a line search in {0, 1, 2, 3, 5}. We found that a large $R_l$ does not always translate to better performance, which could be attributed to the capacity of the synthetic images.
>
> ----
> **Q2: When a class does not appear in a local dataset, what should we expect the synthetic data of that class to look like?**
>
> We would like to clarify that the initialized label set contains only the classes that appear on a client, which are typically not regarded as sensitive to privacy. The images may look random and could be not interpretable for humans. However, we expect that the synthetic images associated with a specific class will also carry information of other classes, thereby making the information class-agnostic.
>
> ----
> **Q3: How do the authors decide the synthetic dataset size of each client? The data complexity and local data distribution should also be considered when deciding the synthetic dataset size.**
>
> Our work considers replacing the plain gradients with synthetic images and explores how to carry more training information given the same amount of communication costs. Therefore, we set the size roughly the same as the model parameters (in Table 1, ours with 0.96$\times$ vs. the baselines with 1$\times$ costs) and evenly assign the size to each class.
>
> On the other hand, we agree with the reviewer that the size could be set according to the distributions and probably even to different classes. However, defining a privacy-preserving (should potentially be data-independent; otherwise, a DP mechanism is required for the chosen size) and reliable metric is non-trivial. We will leave it for future work.
>
> ----
> **Q4: Section 5.1 mentions that the learning rate is 100. Is it a typo?**
>
> No. We notice that the gradients that arrive at the synthetic images are typically small. Therefore, we set it to 100 to approximate the training information in a reasonable time (e.g., due to limited computation on edge devices). We also notice such settings in existing dataset distillation works. For instance (implemented by [1] and the other official codes), gradient matching [2] uses 1, distribution matching [3] uses 10, and MTT [4] uses 10000.
>
> ----
> **Q5: What do the "Fixed, Max, Median, and Min" radius selection mean? Do they matter for private or non-private settings?**
>
> **Recap.** Our method considers two kinds of effective approximation regions, namely $r$ and $r_k$, respectively. The first is a predefined radius that the server expects the client to approximate. However, local data may introduce different difficulties in approximation. Therefore, the clients assess and report the effective approximation region ($r_k$), defined as the radius where the loss values of synthetic data deviate from the real loss (Figure 2).
>
> **The strategies.** After client training, the server will receive a set of radii $\{r_k\}$ suggested by the K clients. The server can decide how to leverage the information. **Fixed** ignores the information and uses the predefined $r$; **Max**, **Median**, and **Min** operate based on how much the server trusts the clients. Our experiments mainly use min, the most conservative strategy. The detailed ablation study is provided in Sec. D.2.
>
> **Strategy in DP settings.** Since the client-suggested radius $r_k$ is data-dependent and could introduce additional privacy concerns, we only use the predefined $r$ in our DP experiments.
>
> ----
> **Q6: When comparing the communication cost, shouldn't we compare the best end-to-end performance with fixed communication costs (e.g., 500MB) by varying the communication rounds and with the best hyper-parameters?**
>
> We have precisely adhered to the setup in this work. We begin with tuning the gradient-sharing baselines and set the size of synthetic sets to match the communication costs, thus creating a constant cost benchmark as suggested by the reviewer. Based on the size, we searched the other hyper-parameters (except for #communication rounds) for our FedLAP.
>
> ----
> **Q7: Does a model with more parameters benefit from or lose advantage with the proposed method?**
>
> The benefits will hold if the approximation quality does not decrease.
>
> ----
> **Q8:  the notations in Appendix (T?) are not consistent with Algorithm 1 (R_b?)**
>
> Our algorithm accesses the data more than $R_b$ times. It requires "$R_b \times R_i \times$ *the number of iterations to achieve the radius $r$*" times. Thus, we use $T$ for conciseness.

---

> > ### Author Response · Authors · 2023-11-16
> > **Reference of the response**
> >
> > **Reference**
> >
> > [1] The offical implementation of "GLaD: Genenralizing Dataset Distillation via Deep Generative Prior," https://github.com/GeorgeCazenavette/glad .
> >
> > [2] Zhao, Bo, Konda Reddy Mopuri, and Hakan Bilen. "Dataset Condensation with Gradient Matching." Ninth International Conference on Learning Representations 2021. 2021.
> >
> > [3] Zhao, Bo, and Hakan Bilen. "Dataset condensation with distribution matching." Proceedings of the IEEE/CVF Winter Conference on Applications of Computer Vision. 2023.
> >
> > [4] Cazenavette, George, et al. "Dataset distillation by matching training trajectories." Proceedings of the IEEE/CVF Conference on Computer Vision and Pattern Recognition. 2022.

---

> ### Author Response · Authors · 2023-11-22
> **Reminder for the end of the discussion phase**
>
> Dear Reviewer,
>
> Since we are approaching the end of the discussion phase, we kindly remind you that you can re-evaluate and adjust your score if our response solves your concerns. We will also be more than happy to answer any further questions and comments.
>
> In addition, we have revised the manuscript according to the reviewers' comments. All revision has been marked in red. Specifically,
>
> 1. (Reviewer SbQA) Algorithm 1 and the corresponding descriptions have been revised for better readability
> 2. (Reviewer XtEq, vGpK) We replace the word "sanitize" with "clip" to explicitly refer to DP operations and rephrase the corresponding sentences.
> 3. (Reviewer vGpK) We add a brief conclusion to Sec. 4.4 to summarize the result of our privacy analysis.

---

> ### Author Response · Authors · 2023-11-23
> **Extension to larger architectures**
>
> To further support our claim in **Q7**, we conducted an additional experiment on CIFAR-10 with VGG11 of size 9231114 floating points. For a fair comparison, we assign 300 synthetic images to each class and keep the other hyper-parameters the same as in our paper. The size of synthetic sets comes from $9231114/(32 \times 32 \times 3) \approx 3000$, where $(32 \times 32 \times3)$ is the size of a synthetic image. The quota of 3000 synthetic images is then evenly assigned to 10 classes, resulting in 300 images per class. Thus, both methods consume the same communication cost for one federated round.
>
> Due to the time limit, we provide a preliminary result against FedAvg and FedProx.
>
> | Epoch  | 1     | 2     | 3     | 4     | 5     | Best |
> |--------|-------|-------|-------|-------|-------|-------|
> | FedAvg | 10.00 | 11.92 | 18.76 | 22.33 | 26.75 | 35.45 |
> | FedProx | 10.00 | 10.00 | 9.43 | 10.00 | 9.97 | 54.89 |
> | Ours   | 12.13 | 28.42 | 36.69 | 40.23 | 46.35 | 58.12 |
>
> The preliminary result shows that our method improves much faster than the plain gradient-sharing method (FedAvg), confirming the effectiveness of our method against non-IID distributions. The faster improvement implies better utility and communication efficiency, making our method favorable concerning limited bandwidth. It is worth noting that larger architectures suffer from non-IID distributions more than the simple ConvNet and waste most of the iterations at the beginning stage.

---

### Official Review · Reviewer_QU6w · 2023-10-31

**Soundness:** 2 fair
**Presentation:** 2 fair
**Contribution:** 2 fair
**Rating:** 3
**Confidence:** 5

**Summary:**

The paper presents an approach called FedLAP-DP for federated learning. In contrast to previous methods that share point-wise local gradients for global model update, the new approach proposes to perform global optimization at the server by leveraing synthetic samples received from clients. Experiments are conducted to show the performance of the proposed method as well as some baseline methods.

**Strengths:**

It is interesting to borrow the idea of Dataset Distillation into federated model training to tackel the non-iid data and privacy issues.

The paper is overall clearly structured and easy to follow.

**Weaknesses:**

Though the idea is interesting, it seems to lack formal guarantees on the approximation achieved for each local synthetic dataset and how approximation level affects the learning performance in theory.

As for the DP side, the explicit trade-off between the privacy loss and the learning utility is also unclear for the proposed method.

In experiment, the performance on different settings with different heterogeneity can be further explored. And since you consider the record-level DP, the setting of privacy budget $\epsilon$ is relatively large even for high privacy regime where $\epsilon$ is set to be 2.79.

**Questions:**

1. Can you provide some formal theoretical guarantees for the proposed algorithm, e.g., approximation of the synthetic data learnt compared with the optimal one, and the learning performance, privacy-utility tradeoffs, so on.

2. Is there any possiblity to extend the proposed method to client-level DP, which is very important in cross-device scenarios such as collaboration between massive IoT devices. If not, then what is the barrier for doing this?

---

> ### Author Response · Authors · 2023-11-16
> **Response to Reviewer QU6w**
>
> We appreciate the reviewer’s insightful comment and concerns. We acknowledge that incorporating theoretical analysis could enhance understanding; however, this does not diminish the contributions our work makes to effective non-IID federated optimization with differential privacy guarantees, which is also supported by the feedback from the other reviewers.
>
> We hope the reviewer will re-evaluate the score in light of the following discussion and prospective extensions of our work.
>
> ----
> **Q1: Lack of formal guarantees on gradient approximation and learning performance**
>
> Theoretical understanding of deep learning algorithms is limited in general, and this applies to dataset distillation as well. We agree that more foundational works along this line is needed, but this is beyond the scope of our contribution.
>
> ----
> **Q2: The explicit trade-off between the privacy loss and the learning utility is also unclear for the proposed method.**
>
> It has been analyzed in Sec. 5-3 and Figure 4.
>
> ----
> **Q3: The privacy budget $\varepsilon = 2.79$ is relatively large even for a high privacy regime.**
>
> Determining an appropriate value for $\varepsilon$ to define "a high privacy regime" is subject to debate. Nonetheless, the range of 2 to 8 for $\varepsilon$ is widely accepted in the existing literature, such as in DP-SGD [1] (but in centralized frameworks). In real-world applications, companies like Apple [2] implement values of 4 and 8 in their Safari browser. Therefore, our setup of $\varepsilon$ is a justified choice for maintaining a high privacy regime while also preserving practical utility.
>
> Moreover, the chosen value does not diminish our contribution, as the proposed FedLAP demonstrates consistent superiority over its counterparts when evaluated with smaller $\varepsilon$.
>
> ----
> **Q4: Is there any possibility to extend the proposed method to client-level DP?**
>
> Yes. Consider a scenario where we have a trusted server, a common assumption in client-level differential privacy. One possible extension is the server groups the synthetic images based on client IDs before server training. During each server training iteration, the server draws batches from every group and calculates the average gradient for each batch. Then, it proceeds to clip the gradients and add noise to the average of the clipped gradients. It’s important to note that the noise scale introduced here is based on the rate of client sampling, which is different from the data-record sampling rate considered in our work.
>
> Lastly, the server uses the processed gradients to update the server model. This step completes a single training cycle. The server continues the process until finishing one communication round, thus ensuring client-level DP.
>
> ----
> **Reference**
>
> [1] Abadi, Martin, et al. "Deep learning with differential privacy." Proceedings of the 2016 ACM SIGSAC conference on computer and communications security. 2016.
>
> [2] https://www.apple.com/privacy/docs/Differential_Privacy_Overview.pdf

---

> ### Author Response · Authors · 2023-11-22
> **Reminder for the end of the discussion phase**
>
> Dear Reviewer,
>
> Since we are approaching the end of the discussion phase, we kindly remind you that you can re-evaluate and adjust your score if our response solves your concerns. We will also be more than happy to answer any further questions and comments.
>
> In addition, we have revised the manuscript according to the reviewers' comments. All revision has been marked in red. Specifically,
>
> 1. (Reviewer SbQA) Algorithm 1 and the corresponding descriptions have been revised for better readability
> 2. (Reviewer XtEq, vGpK) We replace the word "sanitize" with "clip" to explicitly refer to DP operations and rephrase the corresponding sentences.
> 3. (Reviewer vGpK) We add a brief conclusion to Sec. 4.4 to summarize the result of our privacy analysis.

---

> > ### Comment · Reviewer_QU6w · 2023-11-22
> >
> > Q1: Lack of formal guarantees on gradient approximation and learning performance
> >
> > No, you can see much previous work on DP+FL or DP+Deep Learning, although there are no theoretical results on DP+Deep Neural Networks, they also contain theoretical guarantees on convex loss or linear loss. So in my opinion, no theoretical explanation is unacceptable for your problem. Also the theory of DP guarantee also follows the previous paper directly and less interesting.
> >
> > For Large epsilon, yes previous paper uses epsilon=8. However, they are for client level rather than sample level of privacy.
> >
> > There are also many problems and issues with the experiments. The datasets you use are very small. As a experimental paper, you need to use real-world scale data. To the best of my knowledge, the previous paper such as DP-Follow the leader, it uses large scale of data. Thus, there is a lack of experiments. Of course the number of clients is only 5, which is also unreasonable.
> >
> > So, I will not change my score.

---

> ### Author Response · Authors · 2023-11-22
> **Response to the follow-up questions**
>
> We thank the reviewer for the prompt reply! The reviewer seems to be raising new issues that are slightly different from the original review. To address these, we offer the following clarification.
>
> ----
> > you can see much previous work on DP+FL or DP+Deep Learning, although there are no theoretical results on DP+Deep Neural Networks, they also contain theoretical guarantees on convex loss or linear loss. So, in my opinion, no theoretical explanation is unacceptable for your problem.
>
> The original review asked how approximation affects performance in our setting, namely deep learning models. As noted by the reviewer and in our previous response, **there are no (or limited) theoretical results on DP+Deep Neural Networks**.
>
> The reviewer now suggests analyzing simplified models with linear or convex functions, which could be done as those studied in approximation theory. However, to our best knowledge, the efficacy of such methods heavily relies on assumptions and characteristics, which are not comprehensively understood in the context of deep learning. Given the entirely different characteristics, such analysis may not provide practically relevant insights. In contrast, the field typically quantifies approximation errors of unknown functions using empirical metrics, which is precisely what we adopt to learn synthetic images, namely, the cosine distance (and a more common metric MSE in the appendix).
>
> Overall, we still acknowledge the importance of theoretical insight in deep learning but believe that pursuing this direction is non-trivial and beyond our current scope. As an initiative exploration, our work has demonstrated promising results, as acknowledged by other reviewers. We welcome any additional references the reviewer might suggest.
>
> ----
> > For Large epsilon, yes previous paper uses epsilon=8. However, they are for client level rather than sample level of privacy.
>
> The value we used is **a common choice in existing works on record-level DP**. For instance, DP-SGD, the reference we provided in the previous response, considers **record-level DP** and **a range of 2-8**.
>
> In addition, though we acknowledge that a tighter privacy constraint is worth investigating, the usable utility of models is equally important. As reported in Table 2, prior works with $\varepsilon=2.79$ and non-IID data splits deliver around 18% accuracy only, which has been far behind useful (compared to random chance 10% and non-DP settings $\approx$ 75%). Therefore, we believe our setup ($\varepsilon \in [2,10] $) is a reasonable choice, which strikes a good balance between privacy and utility.
>
> ----
> > the experiment settings
>
> We consider a **cross-silo** scenario [1], where relatively few but powerful participants attend. **We invite the reviewer to point out the specific work they refer to for a more direct comparison.**
>
> [1] Kairouz, Peter, et al. "Advances and open problems in federated learning." Foundations and Trends® in Machine Learning 14.1–2 (2021): 1-210.
>
> ----
> We are grateful for the continued dialogue and the opportunity to discuss our work further. We hope these points adequately solve the concerns and clarify the rationale behind our methodological choices.

---

### Official Review · Reviewer_XtEq · 2023-11-01

**Soundness:** 3 good
**Presentation:** 3 good
**Contribution:** 1 poor
**Rating:** 5
**Confidence:** 3

**Summary:**

This paper studies federated learning with data condensation. In order to handle data heterogeneity, instead of sending local updates as in FedAvg, this paper proposes a method of sending synthetic data samples. Experiments show that the proposed method can improve model performance as well as reduce communication. To protect privacy, this paper use Gaussian mechanism to enforce record-level differential privacy.

**Strengths:**

This paper is well written and easy to follow. Extensive experiments are conducted and the results look promising.

**Weaknesses:**

My biggest concern is that the novelty may be limited. Data condensation + FL has been well studied, e.g. [1,2]. In particular, using condensed datasets in FL has been discussed in [1]. The novelty of this work looks limited. It would be highly appreciated if the authors can show improvements over existing works in terms of communication, performance, etc.


[1] Liu, Ping, Xin Yu, and Joey Tianyi Zhou. "Meta knowledge condensation for federated learning." arXiv preprint arXiv:2209.14851 (2022).
[2] Behera, Monik Raj, et al. "Fedsyn: Synthetic data generation using federated learning." arXiv preprint arXiv:2203.05931 (2022).

**Questions:**

For DP, the paper says "we sanitize the gradients derived from real data with the Gaussian mechanism"。 Do you clip the gradients or do something to bound the sensitivity? How will this affect the model utility?

---

> ### Author Response · Authors · 2023-11-16
> **Response to Reviewer XtEq**
>
> We are encouraged that the reviewer find our work easy to follow and promising. We hope our response below can mitigate the reviewer's concern about novelty.
>
> ----
> **Q1: My biggest concern is that the novelty may be limited. Data condensation + FL has been well studied, e.g. [1,2].**
>
> None of the mentioned works investigate privacy-preserving federated learning with DP guarantees. In addition, our FedLAP explicitly leverages approximation quality, i.e., in what region the synthetic data resemble the gradients best, to further boost the performance. These differences outline our novelty and contributions.
>
> ----
> **Q2: It would be highly appreciated if the authors can show improvements over existing works in terms of communication, performance, etc.**
>
> In addition to the non-archival works mentioned by the reviewer, we have compared FedLAP to FedDM, a concurrent work that shares a similar motivation and was published in CVPR 2023. Our method demonstrates superior performance (Sec. 5-2 and Table 1) and provides stronger privacy guarantees (discussed in Sec. 2) by explicitly considering effective approximation quality.
>
> ----
> **Q3: Do you clip the gradients or do something to bound the sensitivity? How will this affect the model utility? "we sanitize the gradients derived from real data with the Gaussian mechanism" looks vague.**
>
> Yes, we clip and add noise to achieve an algorithm with DP guarantees. The process introduces noise and could hurt the model utility. Table 2 and Figure 4 show that our method achieves a better utility-privacy trade-off with clipping and Gaussian noise, i.e., in a DP setting. We will revise the sentence to avoid confusion.

---

> > ### Comment · Reviewer_XtEq · 2023-11-23
> > **Thank you for your reply**
> >
> > Thank authors for the response. I still would like to keep my evaluations. It looks like DP seems to be enforced via plaintext adaptation of Gaussian mechanism and post-processing. FedLAP may introduce a better data generation algorithm, but this requires more experiment validations. I believe it would be helpful to compare with other data condensation/distillation methods as listed in [1,2]. By the way, [1] actually comes from ICLR 2023.

---

> ### Author Response · Authors · 2023-11-22
> **Reminder for the end of the discussion phase**
>
> Dear Reviewer,
>
> Since we are approaching the end of the discussion phase, we kindly remind you that you can re-evaluate and adjust your score if our response solves your concerns. We will also be more than happy to answer any further questions and comments.
>
> In addition, we have revised the manuscript according to the reviewers' comments. All revision has been marked in red. Specifically,
>
> 1. (Reviewer SbQA) Algorithm 1 and the corresponding descriptions have been revised for better readability
> 2. (Reviewer XtEq, vGpK) We replace the word "sanitize" with "clip" to explicitly refer to DP operations and rephrase the corresponding sentences.
> 3. (Reviewer vGpK) We add a brief conclusion to Sec. 4.4 to summarize the result of our privacy analysis.

---

> ### Author Response · Authors · 2023-11-23
> **Follow-up response on the contribution and novelty**
>
> We thank the reviewer for the reply and the time spent reading our response. We would like to follow up on the comments below. While we are approaching the end of the discussion, we remain eager to hear from you and hope you can re-evaluate our work in light of the following contributions.
>
> ----
> - **The contribution of the DP part is non-trivial.**
>
> Chen et al. have shown that a trivial extension of using DP-SGD to learn synthetic images leads to sub-optimal performance. Our work extends it to federated settings and associates it with non-IID scenarios for the first time, providing the same privacy protection as FedAvg with DP guarantees while achieving better utility. This aspect is crucial and has been overlooked by prior works, e.g., the ones suggested by the reviewer.
>
> [1] Dingfan Chen, Raouf Kerkouche, and Mario Fritz. Private set generation with discriminative information. In Advances in Neural Information Processing Systems (NeurIPS), 2022.
>
> ----
> - **Theoretical and empirical results**
>
> In addition to the formulation above, we have provided formal privacy analysis in Sec. A and extensive empirical results in Sec. 5. The former analysis builds a starting point for further fundamental understanding of DP+sample synthesis in FL. The latter can serve as a solid baseline for future non-IID federated optimization and could be of practitioners' interest.

---

> ### Author Response · Authors · 2023-11-23
> **Update on additional related work**
>
> We would like to thank the reviewer again for referring us to the related work published in ICLR 2023 (our first version was released roughly at the same as its arXiv version).
>
> We particularly find it interesting, and many proposed components could be beneficial when integrated into our **non-DP** settings, such as conditional initialization and dynamic weight assignment. However, **any data-dependent operations introduce additional privacy risks, and the privacy protection aspect is missing in the paper**. It remains unclear how to incorporate DP into Liu et al. while our method provides rigorous DP guarantees with proper theoretical analysis, clearly outlining our contribution. We will discuss and empirically compare FedLAP to it in our paper.

---

### Official Review · Reviewer_SbQA · 2023-11-01

**Soundness:** 2 fair
**Presentation:** 2 fair
**Contribution:** 2 fair
**Rating:** 3
**Confidence:** 4

**Summary:**

This paper proposes a novel federated learning (FL) algorithm, namely FedLAP-DP, to address the drawback of bias in global optimization in traditional FL. The approach involves generating synthetic data resembling real data on the client side and substituting local gradients with these synthetic samples during transmission to the central server to approximate the global loss landscape. The central server then iterates using these synthetic samples, thus mitigating the bias in global optimization. Additionally, differential privacy (DP) is employed to protect the privacy of synthetic data of clients. This idea is innovative, and the writing quality is also acceptable.

**Strengths:**

1. The approach proposed in this paper involves generating synthetic data resembling real data on the client side and then leverages the synthetic data to update the local model,  thereby reducing the negative impact of DP on the training.
2. To control the communication cost, the size of the synthetic dataset is much smaller than the real client dataset. Thus, in the local training, the proposed approach leverages a small dataset to update a local model with a satisfactory performance.
3. In addition to applying this approach to the DP setting, it can also address the issue of data heterogeneity in FL.
4. Extensive experimental results demonstrate that FedLAP-DP outperforms the traditional approaches with faster convergence under the different DP settings.

**Weaknesses:**

1. I have a question regarding the generation of synthetic data and the iterations performed by the central server. In the case of non-iid data distribution, are the synthetic samples submitted by the clients to the central server consistent with the original data distribution? If so, referring to Algorithm 1, would there still be bias in the iterations conducted by the central server? Could you please provide a detailed explanation of the algorithm design on how the synthetic data is generated?
2. Please further explain the parameters set that appeared in Sec.5 EXPERIMENT Part 5.1.
3. It would be better to remark on each curve in the experimental figures.  Some notations are not clear.
4. The baselines used in this paper are too simple. Some advanced DP-FL baselines should be included in this paper, such as [1], [2], [3] and etc.

The mentioned references are as follows:
[1] Skellam mixture mechanism: a novel approach to federated learning with differential privacy
[2] Dpis: An enhanced mechanism for differentially private SGD with importance sampling
[3] PrivateFL: Accurate, Differentially Private Federated Learning via Personalized Data Transformation

**Questions:**

The following comments should be addressed.
1. I have a question regarding the generation of synthetic data and the iterations performed by the central server. In the case of non-iid data distribution, are the synthetic samples submitted by the clients to the central server consistent with the original data distribution? If so, referring to Algorithm 1, would there still be bias in the iterations conducted by the central server? Could you please provide a detailed explanation of the algorithm design on how the synthetic data is generated?
2. Some notations are not clear. For example, I cannot see the effect of indexes j and l in Algorithm 1.
3. Please further explain the parameters set that appeared in Sec.5 EXPERIMENT Part 5.1.
4. It would be better to remark on each curve in the experimental figures.
5. Some advanced DP-FL baselines should be included in this paper, such as [1], [2], [3] and etc.

The mentioned references are as follows:
[1] Skellam mixture mechanism: a novel approach to federated learning with differential privacy
[2] Dpis: An enhanced mechanism for differentially private SGD with importance sampling
[3] PrivateFL: Accurate, Differentially Private Federated Learning via Personalized Data Transformation

---

> ### Author Response · Authors · 2023-11-16
> **Response to Reviewer SbQA**
>
> We are thankful for the reviewer’s feedback and the raised concerns. We now address the concerns below and also hope our response will encourage the reviewer to re-evaluate the score in light of the responses provided.
>
> ----
> **Q1: Are the synthetic samples submitted by the clients to the central server consistent with the original data distribution?**
>
> No, we do not target realistic data generation but the critical information necessary for training (i.e., the loss landscape approximation). As the provided visualization in Figure 12 and 13 of the Appendix, the generated images are notably different in appearance from the real data.
>
> ----
> **Q2: If so, referring to Algorithm 1, would there still be bias in the iterations conducted by the central server?**
>
> No. As discussed in the previous response and Sec. 3-1, the bias comes from the biased local optimization and naive weight average. Instead, our method communicates the critical training information and enables unbiased optimization on the server side (Sec. 4-3). However, we also admit that the approximation could affect the performance, which leaves space to improve, as discussed in Sec. 6.
>
> ----
> **Q3: Could you please provide a detailed explanation of the algorithm design on how the synthetic data is generated?**
>
> As shown in Algorithm 1, the process begins with sampling a batch of real data and calculating the mean gradients $g^\mathcal{D}$. These gradients serve as a static objective for training synthetic images, as specified in Eq. 8 and 17 (see Appendix). Subsequently, synthetic images are processed through the identical network to compute their mean gradients. Based on Eq. 8, the synthetic images are then adjusted to align their gradients with $g^\mathcal{D}$. This procedure repeats ​$R_b$ times for every batch of real data. We are readily available to provide further clarification if needed.
>
> ----
> **Q4: The parameter set that appeared in Sec.5 EXPERIMENT Part 5.1.**
>
> Is the reviewer referring to the set ($R_i$ , $R_l$ , $R_b$, $r$)? If so, we offer additional explanations to the descriptions in Algorithm 1 and Sec. 4 as follows.
>
> $R_i$ denotes the number of sampled training trajectories, i.e., the empirical approximation of the expectation in Eq. 7. Within each sampling, $r$ and $R_l$ control the length of the trajectory and step size, respectively. Lastly, $R_b$ controls how many times gradients generated by a real batch will be used to update the synthetic images to deliver the same gradients.
>
> ----
> **Q5: It would be better to remark on each curve in the experimental figures. Some notations are not clear.**
>
> We kindly request that the reviewer provide additional details regarding any specific notations found to be unclear. We believed we had ensured all figures included clear legends and that font sizes were adjusted for easy readability. We would welcome more specific feedback to improve our presentation further.
>
> ----
> **Q6: The baselines used in this paper are too simple. The reviewer suggested three related works to compare.**
>
> [1] and [3] aim at a different problem, making them incomparable to our FedLAP. [1] solves the performance degradation caused by the discretization of multi-party computing (MPC), while our work does not consider MPC. [3] targets personalized federated learning and assumes individual objectives for different clients, while we consider general federated learning and a unified global objective. We have discussed the difference in Sec. 3-1. Concerning the settings, they are incomparable to our FedLAP.
>
> [2] mitigates DP performance degradation by importance sampling. Since it directly works on data, it may be complementary and could be advantageous when integrated into our methodology. However, it does not diminish our contribution to non-IID DP federated optimization based on synthetic image generation.  We will discuss the relevant works in our paper.
>
> **Edit**: We updated the response to clarify the difference and explain why [1] and [3] are incomparable to our method.
>
> ----
> **Q7: The effect of indexes $j$ and $l$ in Algorithm 1.**
>
> We apologize for the confusion and will clarify it in the paper.
>
> The indices were omitted for conciseness. In particular, i and j are related to the update status of synthetic images. It can be denoted as $S^{i, j}_k$ with $i$ being the number of training trajectories that the synthetic images have observed, $j$ the number of updates given a real batch, and $k$ the index of local clients. For $l$, it denotes the number of steps an inner loop will run.

---

> > ### Author Response · Authors · 2023-11-20
> > **Update on Q6**
> >
> > We notice that [3] is also not comparable to our FedLAP.
> >
> > [3] targets personalized federated learning, where each client has a separate test set and individual objectives. It proposes learning a personalized data transformation on each client to mitigate the bias introduced by DP.
> >
> > On the other hand, as described in Sec. 3-1, we consider general federated learning, where all clients have a singular global objective and do not have any assumptions on the further test data distribution. Concerning the entirely different assumption and settings, [3] is not comparable to our FedLAP.

---

> ### Author Response · Authors · 2023-11-22
> **Reminder**
>
> Dear Reviewer,
>
> Since we are approaching the end of the discussion phase, we kindly remind you that you can re-evaluate and adjust your score if our response solves your concerns. We will also be more than happy to answer any further questions and comments.
>
> In addition, we have revised the manuscript according to the reviewers' comments. All revision has been marked in red. Specifically,
>
> 1. (Reviewer SbQA) Algorithm 1 and the corresponding descriptions have been revised for better readability
> 2. (Reviewer XtEq, vGpK) We replace the word "sanitize" with "clip" to explicitly refer to DP operations and rephrase the corresponding sentences.
> 3. (Reviewer vGpK) We add a brief conclusion to Sec. 4.4 to summarize the result of our privacy analysis.

---

### Author Response · Authors · 2023-11-16
**General response**

We sincerely appreciate all reviewers’ time and their valuable feedback. We are pleased to see that the reviewers find our method interesting (QU6w) and effective in improving communication costs (SbQA), non-IID performance (SbQA, QU6w), and privacy utility (SbQA, QU6w), leading to promising performance (SbQA, XtEq, vGpK).

Our proposed FedLAP provides a novel federated approach based on data synthesis with DP guarantees. As appreciated by the majority of the reviewers, it demonstrates promising performance and outlines the contribution of our work.

While the reviewers find our manuscript well-written and easy to follow (XtEq, QU6w), we will clarify further the confusion and address specific concerns in each thread.